# Stepwise introduction of three different transition metals in metallo-supramolecular polymer for quad-color electrochromism

Manas Kumar Bera [1], Yoshikazu Ninomiya[1] & Masayoshi Higuchi [1✉]

Metallo-supramolecular polymers (MSPs) show unique electrochemical and optical properties, that are different to organic polymers, caused by electronic interactions between metals and ligands. For the development of quad-color electrochromic materials, here we report the stepwise introduction of three different transition metal ions into an MSP, utilizing the different complexation abilities of the transition metals. An MSP with Os(II), Ru(II), and Fe(II) (polyOsRuFe) was synthesized via a stepwise synthetic route through the formation of an Os(II) complex first, followed by the introduction of Ru(II) to the Os(II) complex, and finally the attachment of Fe(II) to the Os(II)-Ru(II) complex to produce the polymer. This synthetic procedure was extended to fabricate MSPs that comprised Co(II)/Ru(II)/Os(II) and Zn(II)/Ru(II)/Os(II). The synthesized MSPs showed a broad optical and electrochemical window due to the coupling of three heterometallic segments into the polymer. Introducing acetate anion as the counter anion greatly enhanced the solubility of polyOsRuFe in methanol. A thin film of polyOsRuFe was prepared on ITO/glass by spin-coating the methanol solution, and its reversible quad-color electrochromism was demonstrated.

[1] Electronic Functional Macromolecules Group, Research Center for Functional Materials, National Institute for Materials Science (NIMS), Tsukuba Ibaraki, Japan. ✉email: HIGUCHI.Masayoshi@nims.go.jp

Metallo-supramolecular polymers (MSPs), which are synthesized by a 1:1 complexation of metal ions and a ditopic ligand, have attracted considerable attention for a wide range of applications, including electrochromic (EC) displays, memory devices, sensors, energy storage devices, and anticancer therapies[1–7]. The electronic interactions between the metal and ligand in the polymer chains cause unique electrochemical, optical, emissive properties, unlike the conventional organic polymers. Generally, one metal species is included in MSPs, but the introduction of two metal ion species in the polymer chain has attracted increasing research attention because the coupling of dual metal species in the polymer is expected to expand the functions of MSPs[2,8–17]. Stang et al. developed Pt/Zn-based heteroMSP[8]. The current authors reported Os/Fe-based heteroMSP[10]. Li et al. developed a Ru/Fe-based heteroMSP and visualized the polymer chain directly[18]. However, there are no reports on the control of three metal ion species to the best of the authors' knowledge.

The reaction of a metal complex with an organic compound is useful for forming a metal-containing ligand that can undergo further complexation with heterometal ions to give heterometallic supramolecular complexes or polymers. Thus, it was assumed that a similar strategy could organize three heterometal ions into MSP. However, the realization of such systems is challenging because of the different binding abilities of heterometal ions as well as the different stabilities of the corresponding heterometal complexes, which makes it difficult to organize three homoleptic heterometal complexes into a metallo-supramolecular chain. Although the fabrication of various structural heterotrimetallic (and multimetallic) supramacromolecules (discrete architectures) has been explored widely by developing either one-pot or stepwise multicomponent self-assembly processes[19–31], there has been little interest in the construction of three heterometal ion-decorated MSP, i.e., heterotrimetallic–supramolecular polymer (HTMSP).

Given this background and inspired by the fascinating features of the recently reported heteroMSP, this paper reports the synthesis of HTMSP by introducing three heterometal ions [Fe(II)/Ru(II)/Os(II)] into a linear MSP chain (called polyOsRuFe). The heterometal ions in polyOsRuFe were introduced in homoleptic coordination environments made from two 2,2′:6′,2″-terpyridine (tpy) units. A stepwise synthetic route was designed for the synthesis of polyOsRuFe, involving the stepwise introduction of a strong coordination metal ion, Os(II), followed by another strong coordination metal ion, Ru(II), and then by a weak coordination metal ion, Fe(II). Strong coordination refers to the binding strength of the tpy-M(II)-tpy connectivity (where M = Ru or Os or Fe). Schubert et al. reported that the binding strength of tpy-Ru(II)-tpy connectivity is higher than that of tpy-Fe(II)-tpy connectivity[32]. Newkome et al. showed that the binding strength of tpy-Os(II)-tpy and tpy-Ru(II)-tpy connectivity were higher than that of tpy-Fe(II)-tpy connectivity[33]. These studies confirmed that Os(II) and Ru(II) form strong coordination with tpy compared with Fe(II). In this context, Os(II) or Os(II)-Ru(II) containing supramolecular systems (polymers or supramacromolecules) have received less attention owing to the harsh reaction conditions and low yields associated with the Os(II) ion[14,33–35]. In the present study, a synthetic route was designed and established to construct polyOsRuFe for obtaining a significant product yield (%) in each step. The metal ion could be varied in the third step of the designed synthetic route to produce HTMSPs with the Co/Ru/Os and Zn/Ru/Os sequence. The HTMSPs exhibited a broad optical and electrochemical window, which originated from the combination of three heterometallic segments into the linear MSP chain. The processability of polyOsRuFe was enhanced by tuning its solubility in various high and low boiling solvents, including green solvents, such as EtOH and H$_2$O, by adjusting the counteranions. A thin film of polyOsRuFe was prepared on ITO/glass. The film exhibited quad-color electrochromism upon the stepwise oxidation of the three heterometal ions, showing great potential for the development of voltage-tunable multicolor EC displays[4].

## Results

**Synthesis of polyOsRuFe.** An MSP with the Fe(II)/Ru(II)/Os(II) sequence (polyOsRuFe) was synthesized using 2,2′:6′,2″-terpyridine (tpy) as the coordinating ligand. tpy was chosen because of its strong coordination ability toward various metal ions through the tpy-M(II)-tpy connectivity, which is either labile or nonlabile in nature[5,19,36]. Here, tpy-metal chemistry was used to decorate the three heterometal ions into a linear MSP chain. Although MSPs are typically synthesized by mixing a tpy containing ditopic ligand and metal ion to achieve coordination-driven self-assembly[3–6,35], this strategy could not be used to decorate three heterometal ions at one time into a linear MSP chain. This is because direct mixing of a ditopic ligand and three heterometal ions could result in random polymers (a mixture of different homometallic and heterometallic supramolecular polymers). In addition, various metal complexes (Supplementary Fig. 1a–b) can be formed because of the variable reaction conditions for the coordination complexation of tpy toward heterometal ions and the different strengths of tpy-M(II)-tpy connectivity[19,36]. For example, tpy can generally bind Os(II) at high temperatures and Ru(II) at moderate to high temperatures with strong tpy-M(II)-tpy connectivity for both. By contrast, binding of Fe(II) occurs at moderate to room temperature (25 °C) with comparatively weak tpy-M(II)-tpy connectivity[9,14,19,33,35]. Thus, after careful consideration of the reaction conditions of Os(II), Ru(II), and Fe(II) with tpy and the stability of the tpy-M(II)-tpy connectivity, a stepwise synthetic route was designed and developed for the precise synthesis of polyOsRuFe (Supplementary Fig. 1c) through the stepwise harnessing of first a strong coordination metal ion, Os(II), followed by another strong coordination metal ion, Ru(II), and then, a weak coordination metal ion, Fe(II) (Fig. 1).

As shown in Fig. 1, compound 2 was prepared from 4-bromo-2,2′:6′,2″-terpyridine (compound 1) and (NH$_4$)$_2$OsCl$_6$. This was followed by the Suzuki coupling of compound 2 with compound 3 to produce TOsT in 73% yield[14]. The TOsT is a ditopic ligand of Os(II) complex with two free (uncoordinated) tpy units. Ru(II) was then attached to one side of TOsT by refluxing TOsT with compound 4 in a mixture solvent of CHCl$_3$/CH$_3$OH (1:1, v/v) at 75 °C for 15 h to obtain an asymmetric intermediate compound (TOsRuBr) in 44% yield, which is the key step of this synthetic route for the successful introduction of three heterometal ions into an MSP chain. TOsRuBr was separated by column chromatography as the first fraction. After the first fraction, the second fraction of undesired product appeared, which was probably the product of both the side attachment of compound 4 with TOsT. Finally, the Suzuki coupling of TOsRuBr with compound 3 gave rise to the modified ditopic ligand (TOsRuT) containing both Os(II) and Ru(II) with two free tpy units in opposite directions, which can undergo further complexation with another metal ion to make a linear MSP. The typical synthesis of metallo-supramacromolecules (discrete architectures) and MSPs is mostly achieved via precipitation of the intermediate or final compounds by making PF$_6^-$ as the counteranions through the direct addition of a PF$_6^-$ anion-containing salt to the reaction mixture. This makes it easier to isolate and characterize the compounds in CH$_3$CN media. This strategy was avoided in the designed synthetic route (Fig. 1) when preparing intermediate compounds and final polymers. This is because once the

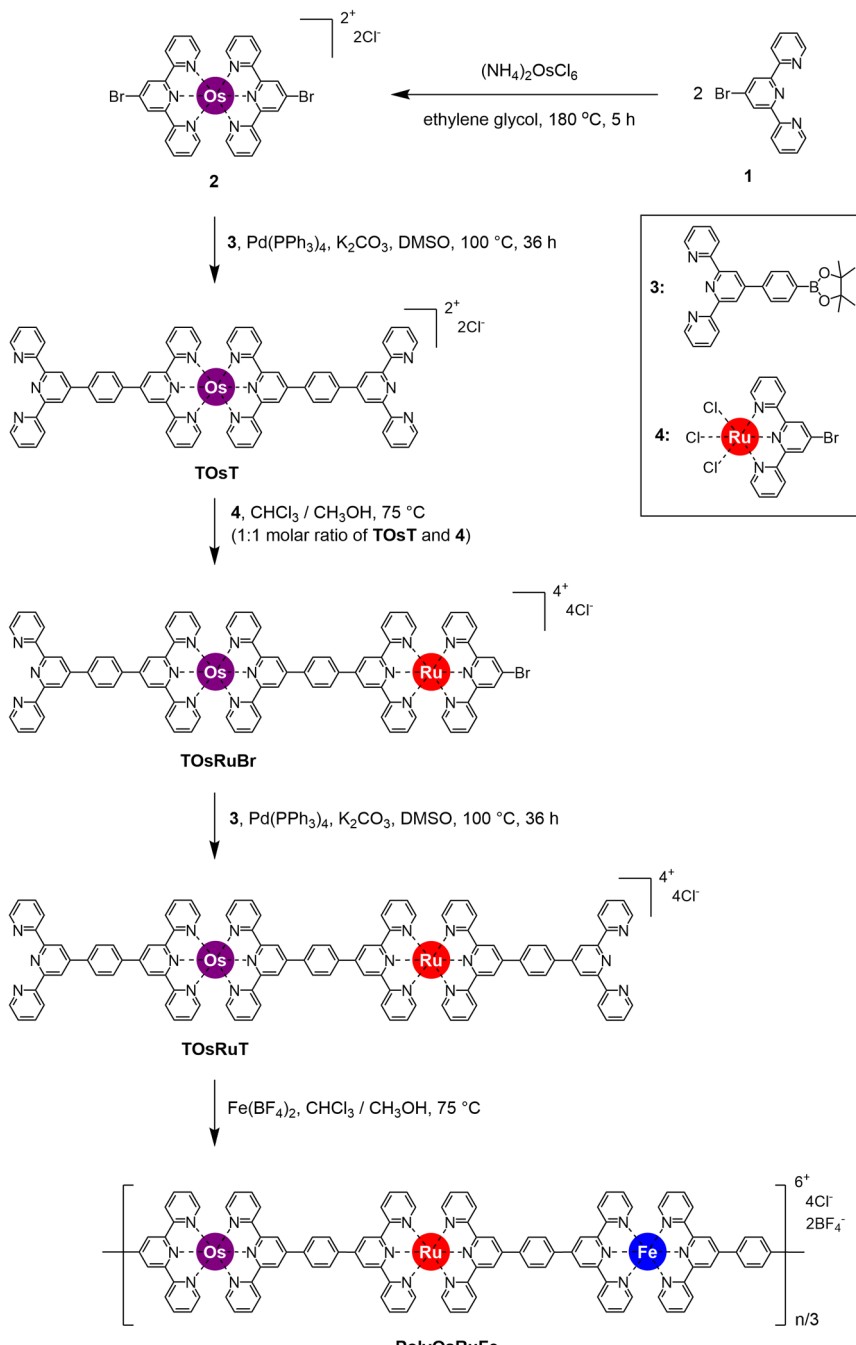

**Fig. 1 Synthesis and chemical structure of polyOsRuFe.** The stepwise synthesis of polyOsRuFe.

counteranions of a coordination complex are converted to $PF_6^-$, the solubility of the complex is limited in few solvents. In addition, the replacement of large $PF_6^-$ counteranions of the complex is difficult using other small-size counteranions, such as $Cl^-$, which helps to dissolve the compound in other solvents apart from $CH_3CN$ (more about this strategy is shown experimentally in a later section). Thus, the modified ditopic ligand TOsRuT was synthesized with $Cl^-$ as the counteranion. The synthesis of TOsRuT was also targeted using other approaches, but they were unsuccessful (see Supplementary Fig. 2 for the details of the other approaches). Newkome et al. reported that the binding strength of tpy-M(II)-tpy connectivity (M = Ru or Os or Fe) followed the order Ru > Os > Fe[33]. This order of binding strength of the metal complex suggests that TOsRuT

could be synthesized in two ways because both the metal ions form complexes at high temperatures compared with Fe(II): either the first complexation of Os(II) followed by the complexation of Ru(II) or the first complexation of Ru(II) followed by the complexation of Os(II). Conversely, approaches to obtaining TOsRuT were unsuccessful, as shown in Supplementary Fig. 2. However, the synthesis of TOsRuT with $Cl^-$ as counteranions was finally successful by developing the synthetic route (Supplementary Fig. 3) shown in Fig. 1. The Supplementary methods outline the synthetic procedure and characterization of the compounds (Supplementary Figs. 4–18). Following this synthetic route shown in Fig. 1, undesirable side product formation associated with Os(II) was avoided and a high product yield was obtained in each step.

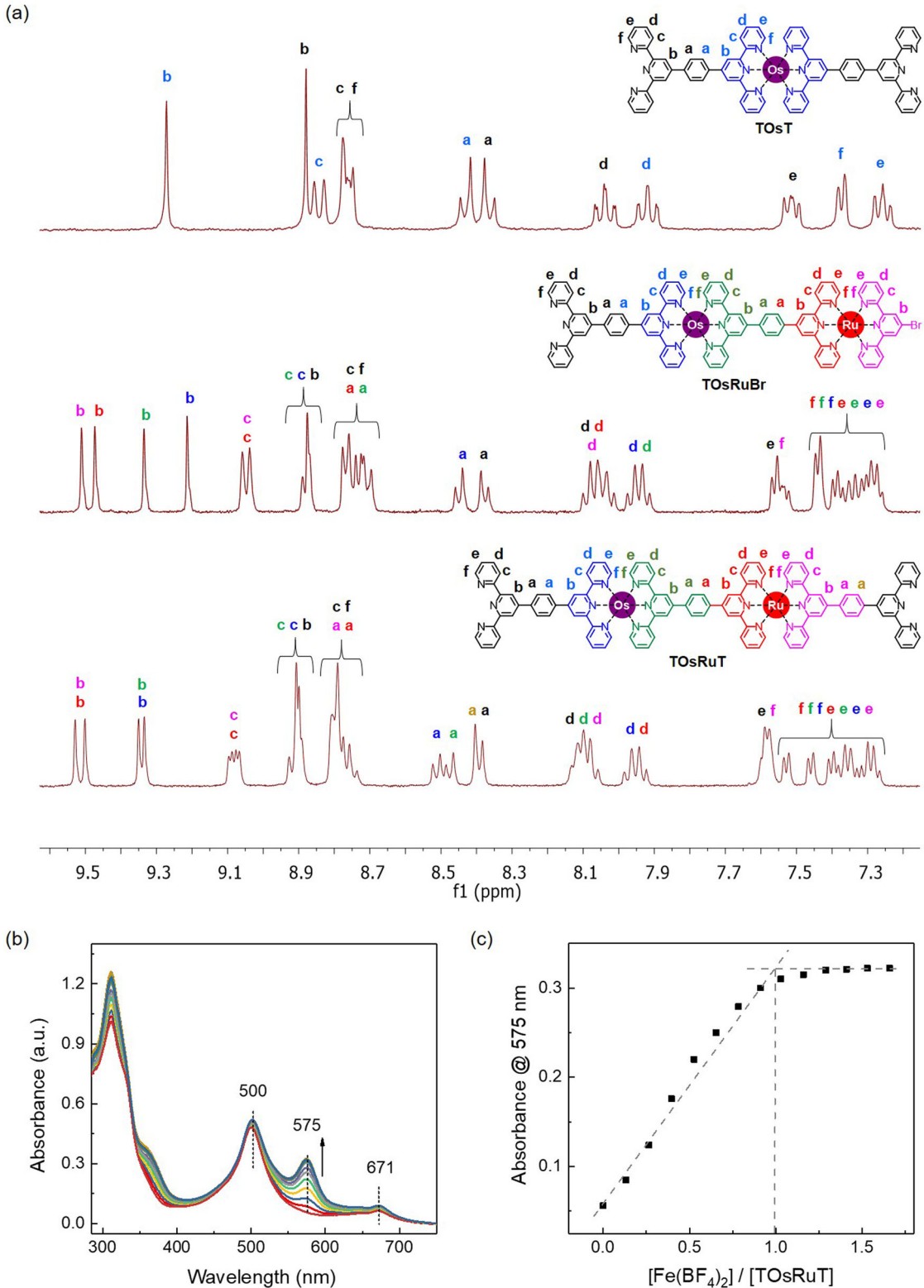

**Fig. 2 Characterization of the intermediate compounds by NMR and titration of TOsRuT with Fe(II). a** Comparison of the $^1$H NMR spectra of TOsT, TOsRuBr, and TOsRuT with peak assignments in CD$_2$Cl$_2$/CD$_3$OD (1:1, v/v). Assignment of protons: b = 3′,5′; c = 3,3″; d = 4,4″; e = 5,5″; f = 6,6″. **b** Change in UV–vis absorption upon the stepwise addition of Fe(BF$_4$)$_2$ in CH$_3$OH to TOsRuT (5 × 10$^{-6}$ M in CH$_2$Cl$_2$/CH$_3$OH; 1:1, v/v). **c** Plot of absorbance of TOsRuT at 575 nm as a function of the [Fe(BF$_4$)$_2$]/[TOsRuT] ratio.

Figure 2a shows the $^1$H nuclear magnetic resonance (NMR) spectra of TOsT, TOsRuBr, and TOsRuT with readily assignable peaks based on the two-dimensional NMR ($^1$H–$^1$H correlation spectroscopy (COSY) and nuclear overhauser effect spectroscopy

(NOESY)) spectra. They clearly show the gradual attachment of the tpy-metal segment upon the stepwise synthesis of the intermediate compounds (from TOsT to TOsRuBr and to TOsRuT). In addition, moving from TOsT to TOsRuBr,

(a)

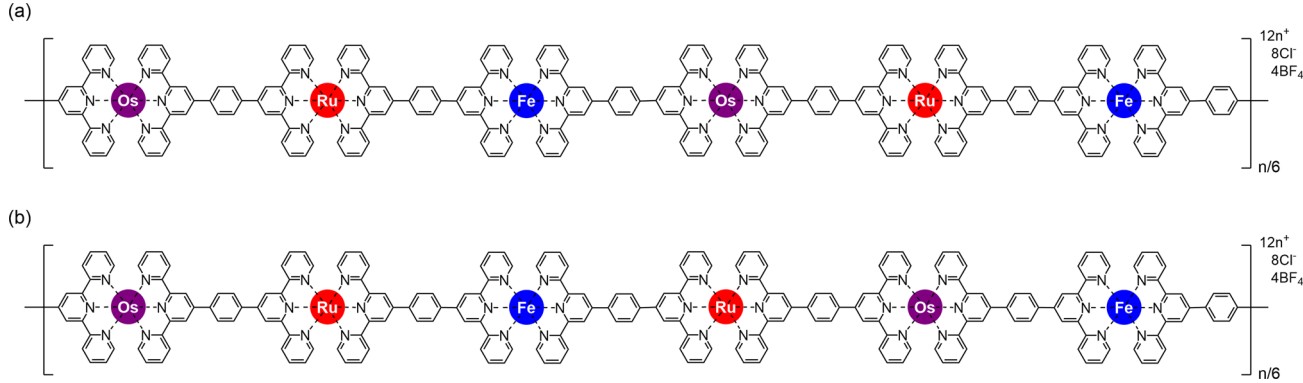

(b)

**Fig. 3 Possible chemical structures of polyOsRuFe. a** Head-to-tail structure and **b** head-to-head and tail-to-tail structure of polyOsRuFe.

additional 3′,5′ resonances ($\delta$ = 9.52, 9.48, and 9.22) appeared, which were shifted downfield ($\delta$ = 9.53, 9.50, and 9.33) in TOsRuT. The presence of redox-active Os(II) and Ru(II) in TOsRuT was also confirmed by cyclic voltammetry (CV), which revealed two distinct redox potentials for the Os(II)/Os(III) and Ru(II)/Ru(III) redox pair with $E_{1/2}$ of 0.63 and 0.99 V, respectively (Supplementary Fig. 18b). The TOsRuT contains two free tpy units in opposite directions (similar to a ditopic ligand), which allows it to undergo further complexation with another different metal ion.

The complexation behavior of TOsRuT ($5 \times 10^{-6}$ M in $CH_2Cl_2/CH_3OH$; 1:1, v/v) with Fe(II) was monitored by the UV–vis spectral change with the gradual addition of Fe(II) [as Fe $(BF_4)_2$ $6H_2O$ salt in $CH_3OH$] at room temperature (25 °C) (Fig. 2b). The TOsRuT exhibited three absorption bands: 311 nm for the $\pi$–$\pi^*$ transition, 500 nm for the metal-to-ligand charge-transfer (MLCT) bands of <tpy-Ru(II)-tpy> connectivity + singlet MLCT for <tpy-Os(II)-tpy> connectivity, and 671 nm for the triplet MLCT for <tpy-Os(II)-tpy> connectivity (Supplementary Fig. 18a). Upon the gradual addition of Fe(II) to the solution of TOsRuT, a new absorption band appeared at 575 nm corresponding to the MLCT absorption band for <tpy-Fe(II)-tpy> connectivity (Fig. 2b). With increasing Fe(II) concentration, the intensity of the MLCT band at 575 nm increased gradually and was finally saturated with a molar ratio close to 1:1 for TOsRuT: $Fe(BF_4)_2$ (Fig. 2c). Such complexation behavior of TOsRuT with Fe(II) suggests that TOsRuT can bind another metal ion to form HTMSP. Further increases in the Fe(II) concentration into the complexed mixture did not affect the intensity of the 575 nm MLCT band, suggesting that the polymer was stable in solution (the polymer was precipitated when the solution mixture was reached saturation).

Finally, the polyOsRuFe was synthesized by complexation of TOsRuT with $Fe(BF_4)_2$ $6H_2O$ (1:1 molar ratio of ligand and metal) in a mixed solvent of $CHCl_3$ and $CH_3OH$ (1:1, v/v) at 75 °C for 24 h, which gave the final product as a precipitate in 90% yield. The polyOsRuFe was mainly soluble in high boiling solvents, such as DMSO and DMF. The polymer was characterized by $^1H$ NMR, Fourier transform infrared spectroscopy (FTIR), X-ray photoelectron spectroscopy (XPS), UV–vis absorption spectroscopy, and CV. The $^1H$ NMR spectra of polyOsRuFe revealed significant peak broadening compared with TOsRuT, confirming the formation of the polymer. In addition, the 3′,5′ peaks of free tpy units in TOsRuT were shifted further to a lower field in polyOsRuFe because of the complexation of free tpy units with Fe(II) (Supplementary Figs. 19–20). The molecular weight ($M_w$) of the polyOsRuFe was estimated to be a high value of $5.44 \times 10^6$ Da using the right angle light scattering (RALLS)

method. Such a high molecular weight polymer is associated with the formation of molecular aggregates in solution, which is commonly observed for MSPs[8,14].

FTIR spectroscopy was conducted to obtain further confirmation on the formation of intermediate compounds (TOsT, TOsRuBr, and TOsRuT) and polyOsRuFe. The FTIR spectra revealed two types of C=C stretching vibrations in TOsT, TOsRuBr, and TOsRuT, as these compounds have both free (uncoordinated) and coordinated tpy units. The C=C stretching frequencies were at 1583 cm$^{-1}$ for the free tpy units and ≥1600 cm$^{-1}$ for the coordinated tpy units. However, the polyOsRuFe exhibited only the C=C stretching frequencies of coordinated tpy units at 1604 cm$^{-1}$ (Supplementary Fig. 21), suggesting the formation of a polymer.

A closer look at the chemical structure of the modified ditopic ligand TOsRuT suggests that an asymmetric structure can be considered for this ligand because of the presence of one Os(II) and one Ru(II) complex in the modified ditopic ligand. Suppose that the reactivity of two tpy units at the two ends of TOsRuT is different because of the presence of two heterometal ions. The TOsRuT can be imagined as a structure with one side as the head and another as the tail. Hence, when TOsRuT reacts with Fe(II) to make the polymer, it could be anticipated that the resulting polymer (polyOsRuFe) may be formed with a regular head-to-tail structure (Fig. 3a) or a regular structure with some irregular head-to-head/tail-to-tail structures (Fig. 3b). For example, if the Ru(II) side binds Fe(II) better than the Os(II) side, the first step of the assembly process would be the formation of OsRuFeRuOs dimers, which would lead to an ordered (OsRuFeRuOs-Fe-OsRuFeRuOs)$_n$ polymeric sequence. Conversely, if the reactivity of two tpy units at the two ends of TOsRuT is the same, the complexation with Fe would lead to polyOsRuFe with a completely random sequence. Presently, there is no experimental evidence to support this assumption (theoretical prediction). More investigation in this direction is currently underway.

XPS confirmed the formation of polyOsRuFe. The XP spectrum showed the characteristic peaks for the binding energy of Fe 2p at 707.9 and 720.6 eV, Ru 3d at 280.2 eV, Os 4f at 52.9 and 50.2 eV, and N 1s at 399.2 eV (Fig. 4a). These observations indicate the presence of three divalent heterometal ions [Fe(II)/Ru(II)/Os(II)] in polyOsRuFe.

The thermal stability of polyOsRuFe was examined by thermogravimetric analysis (TGA). TGA analysis revealed two degradation points. The first point at ~365 °C was assigned to the breaking of long chains. The second point (>700 °C) was attributed to the breaking of the ligand backbone (Supplementary Fig. 22)[14], suggesting the high thermal stability of the polyOsRuFe.

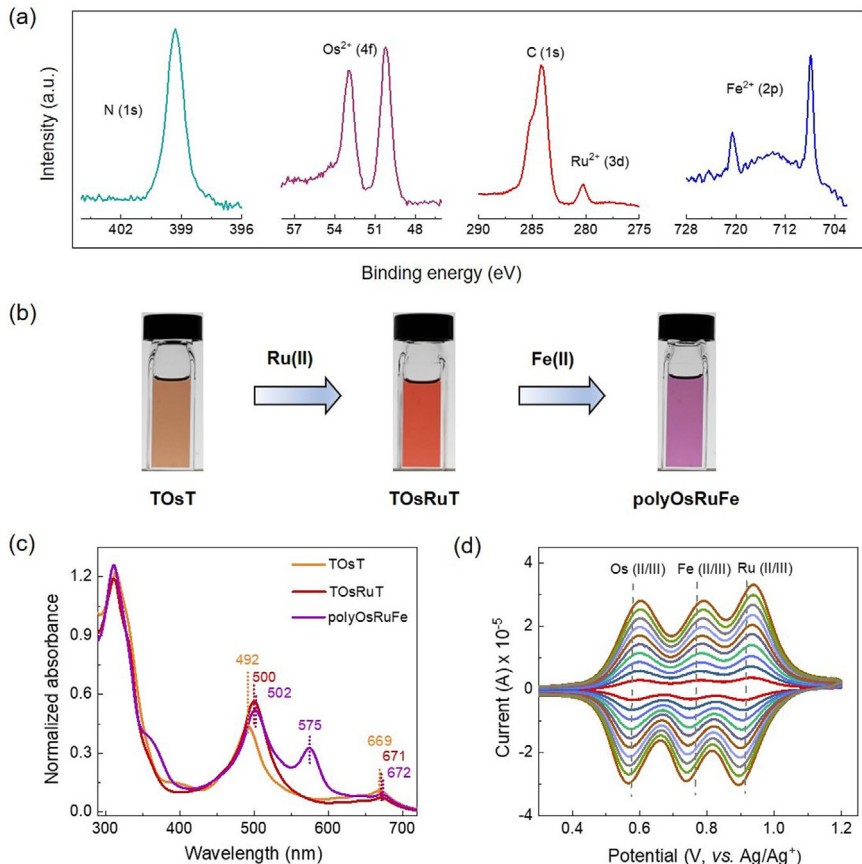

**Fig. 4 XPS, UV–vis, and CV analysis of polyOsRuFe. a** Normalized X-ray photoelectron spectrum of polyOsRuFe showing N (1s), $Os^{2+}$ (4f), $Ru^{2+}$ (3d), and $Fe^{2+}$ (2p) bands. **b** Change in color in the solution state upon the stepwise complexation of Os(II) in TOsT, followed by Ru(II) in TOsRuT and Fe(II) in polyOsRuFe. **c** UV–vis spectra of TOsT, TOsRuT (both solutions were $5 \times 10^{-6}$ M in $CH_2Cl_2/CH_3OH$; 1/1, v/v), and polyOsRuFe ($5 \times 10^{-6}$ M in DMSO). **d** Cyclic voltammograms of polyOsRuFe at scan rates of 0.01–0.1 V/s in a three-electrode system, electrolyte: 0.1 M $LiClO_4$ in $CH_3CN$.

**Optical properties of TOsT, TOsRuT, and polyOsRuFe**. The optical properties of TOsT ($5 \times 10^{-6}$ M in $CH_2Cl_2/CH_3OH$; 1/1, v/v), TOsRuT ($5 \times 10^{-6}$ M in $CH_2Cl_2/CH_3OH$; 1/1, v/v), and polyOsRuFe ($5 \times 10^{-6}$ M in DMSO) were investigated by observing their visual color in solution and measuring the UV–vis absorption of the solution. The stepwise complexation of the three heterometal ions [first Os(II) and then Ru(II), followed by Fe(II)] was also confirmed by observing the visual color of solutions of the intermediate compounds and polymers (moving from TOsT to TOsRuT and to polyOsRuFe) dissolved in an appropriate solvent. The compound TOsT containing Os(II) complex showed a deep yellow color, which turned red upon complexation with Ru(II) to form TOsRuT. Finally, the complexation of Fe(II) with TOsRuT to produce polyOsRuFe turned the color to violet (Fig. 4b). The UV-spectrum of polyOsRuFe is shown in Supplementary Fig. 23a. Figure 4c presents the UV–vis spectra of TOsT, TOsRuT, and polyOsRuFe. Supplementary Table 1 lists the optical data. The TOsT exhibited a broad absorption window ranging from 314 to 669 nm, including a π–π* transition at 314 nm and a singlet and triplet MLCT absorption at 492 and 669 nm, respectively for <tpy-Os(II)-tpy> connectivity[35,37]. Upon the attachment of Ru(II) to TOsT (in TOsRuT), the MLCT of <tpy-Ru(II)-tpy> connectivity and singlet MLCT of <tpy-Os(II)-tpy> connectivity overlapped. Thus, a broad peak was observed in TOsRuT at 500 nm with a redshift of 8 nm compared with singlet MLCT for <tpy-Os(II)-tpy> connectivity in TOsT. Moreover, triplet MLCT for <tpy-Os(II)-tpy> connectivity in TOsRuT was observed at 671 nm. Upon the

complexation of TOsRuT with Fe(II) (in polyOsRuFe), an additional MLCT band for the Fe(II) complex appeared at 575 nm. Hence, polyOsRuFe displayed a broad absorption window from 311 to 671 nm, including a π–π* transition, MLCT of <tpy-Fe(II)-tpy> connectivity, MLCT of <tpy-Ru(II)-tpy> connectivity, and singlet and triplet MLCT of <tpy-Os(II)-tpy> connectivity.

**Electrochemical properties of polyOsRuFe**. Three redox-active heterometal ions [Fe(II)/Ru(II)/Os(II)] were introduced in polyOsRuFe, and the electrochemical property of the polymer was examined. The electrochemical properties were investigated by CV using a three-electrode system (glassy carbon electrode containing the sample as the working electrode (WE), a platinum flag as the counter electrode (CE), and $Ag/Ag^+$ as the reference electrode, electrolyte: 0.1 M $LiClO_4$ in $CH_3CN$) and the electrochemical data are summarized in Supplementary Table 2. The polyOsRuFe exhibited three distinct reversible one-electron redox processes of M(II/III); M = Os, Fe, and Ru with a half-wave redox potential ($E_{1/2}$) of 0.58, 0.76, and 0.92 V for Os, Fe, and Ru, respectively (Fig. 4d). The observed redox potential of the three heterometal ions in polyOsRuFe was comparable with the previously reported heterometallic supramolecular complexes and polymers (see Supplementary Table 3 for the details of the comparison). The scan-rate-dependent (0.01−0.1 V/s) CV study of polyOsRuFe revealed the linear proportionality of the peak current with the scan rate (Fig. 4d and Supplementary Fig. 23b), suggesting a surface-confined electrochemical redox process that is not restricted by slow electrolyte diffusion[10]. Notably,

**Fig. 5 Synthesis and chemical structure of polyOsRuCo and polyOsRuZn.** Synthetic route to polyOsRuCo and polyOsRuZn from TOsRuT.

combining three heterometal complexes into a linear MSP chain produced significant enlargement of the electrochemical window, which could be of interest for potential applications.

**Synthesis of MSPs with Co(II)/Ru(II)/Os(II) and Zn(II)/Ru (II)/Os(II) sequence (polyOsRuCo and polyOsRuZn).** Following the similar polymerization process for polyOsRuFe, the polyOsRuCo and polyOsRuZn were prepared from TOsRuT using $Co(BF_4)_2$ $6H_2O$ and $Zn(ClO_4)_2$ $6H_2O$, respectively (Fig. 5). The details of the synthesis and characterization are given in Supplementary Methods (Supplementary Figs. 24–29). The polymers were isolated as a precipitate from the reaction mixture, filtered, washed with fresh solvent, and dried under vacuum. The polymers were mainly soluble in DMSO and DMF. The $^1H$ NMR spectra of polyOsRuCo and polyOsRuZn revealed the broadening of the peak as like polyOsRuFe compared with TOsRuT with a lower field shift of 3′,5′ peaks of free tpy units in TOsRuT upon polymerization (Supplementary Figs. 24 and 27). The molecular weights for polyOsRuCo and polyOsRuZn measured using RALLS were $1.34 \times 10^7$ and $1.35 \times 10^7$ Da, respectively. XPS revealed characteristic peaks for Co 2p of polyOsRuCo at 780.9 and 798.1 eV and Zn 2p of polyOsRuZn at 1021.3 eV. Both polymers displayed the corresponding peaks for the binding energies of Ru 3d, Os 4f, and N 1s orbitals (Supplementary Figs. 25 and 28). The formation of polyOsRuCo and polyOsRuZn was confirmed by FTIR spectroscopy, which revealed the presence of the C=C stretching frequency of coordinated tpy units at around 1603 $cm^{-1}$ for both polymers (Supplementary Fig. 21). TGA analysis of polyOsRuCo and polyOsRuZn also showed two degradation points: the first point between 270 and 330 °C for breaking of the long chain and the second point (>700 °C) for breaking of the ligand backbone (Supplementary Fig. 22).

**Optical and electrochemical properties of polyOsRuCo and polyOsRuZn.** The optical and electrochemical properties of polyOsRuCo and polyOsRuZn were measured under similar conditions to polyOsRuFe. Supplementary Tables 1 and 2 provide a summary of the optical and electrochemical data. The UV–vis analysis of polyOsRuCo ($5 \times 10^{-6}$ M in DMSO) and polyOsRuZn ($5 \times 10^{-6}$ M in DMSO) also showed a broad absorption window that typically lies in the visible region (309–671 nm). The UV–vis spectra of the polymers included a π–π* transition, singlet and

triplet MLCT of <tpy-Os(II)-tpy> connectivity, and the MLCT of <tpy-Ru(II)-tpy> connectivity (Supplementary Figs. 26a and 29a). The electrochemical properties of polyOsRuCo and polyOsRuZn were measured by CV in a three-electrode system (glassy carbon electrode containing the sample as the WE, a platinum flag as the CE, Ag/Ag+ as the reference electrode). The polyOsRuCo showed three distinct reversible one-electron redox processes of Os(II/III), Ru(II/III), and Co(II/I) with $E_{1/2}$ of 0.58, 0.91, and −1.29 V, respectively (Supplementary Fig. 26b). Interestingly, polyOsRuFe (discussed earlier) and polyOsRuCo are the first examples of MSP where three redox-active metal ions have been integrated together in a homoleptic environment. As Zn(II) complex does not show redox activity, polyOsRuZn exhibits only two reversible redox characteristics corresponding to the Os(II/III) and Ru(II/III) redox process (Supplementary Fig. 29b).

**Synthesis of polyOsRuFe with different counteranions (polyOsRuFe-A, where A = BF₄, Cl, PF₆, and OAc).** Although HTMSPs (polyOsRuFe, polyOsRuCo, and polyOsRuZn) were synthesized, they were mainly soluble in high boiling point solvents DMSO and DMF. This makes it difficult to process the polymers for several applications, particularly in device-related applications where the preparation of a thin film on a substrate is important. Thus, making solution-processable HTMSPs, particularly in low boiling solvents, is another challenge with these heterometallic supramolecular polymers[10]. The counteranions management strategy was examined using polyOsRuFe as an example to tune its solubility in various low boiling solvents and enhance the processability of the polymer. For this purpose, polyOsRuFe with different counteranions (called polyOsRuFe-A, where A = $BF_4^-$, $Cl^-$, $PF_6^-$, and $AcO^-$) were prepared from TOsRuT using various Fe(II) salts, as shown schematically in Fig. 6. The details of the synthesis of polyOsRuFe-As are reported in Supplementary methods (Supplementary Figs. 30–32). All polyOsRuFe-As (A = Cl, PF₆, and OAc) exhibited identical optical and electrochemical behavior as polyOsRuFe (polyOsRuFe-BF₄). Supplementary Tables 4 and 5 summarize the optical and electrochemical properties of polyOsRuFe-A polymers. The cycling stability of polyOsRuFe-OAc was examined (as this polymer was used for EC applications shown in a later section) for 1000 cycles to check the redox stability of polyOsRuFe; the data are shown in Supplementary Fig. 32d.

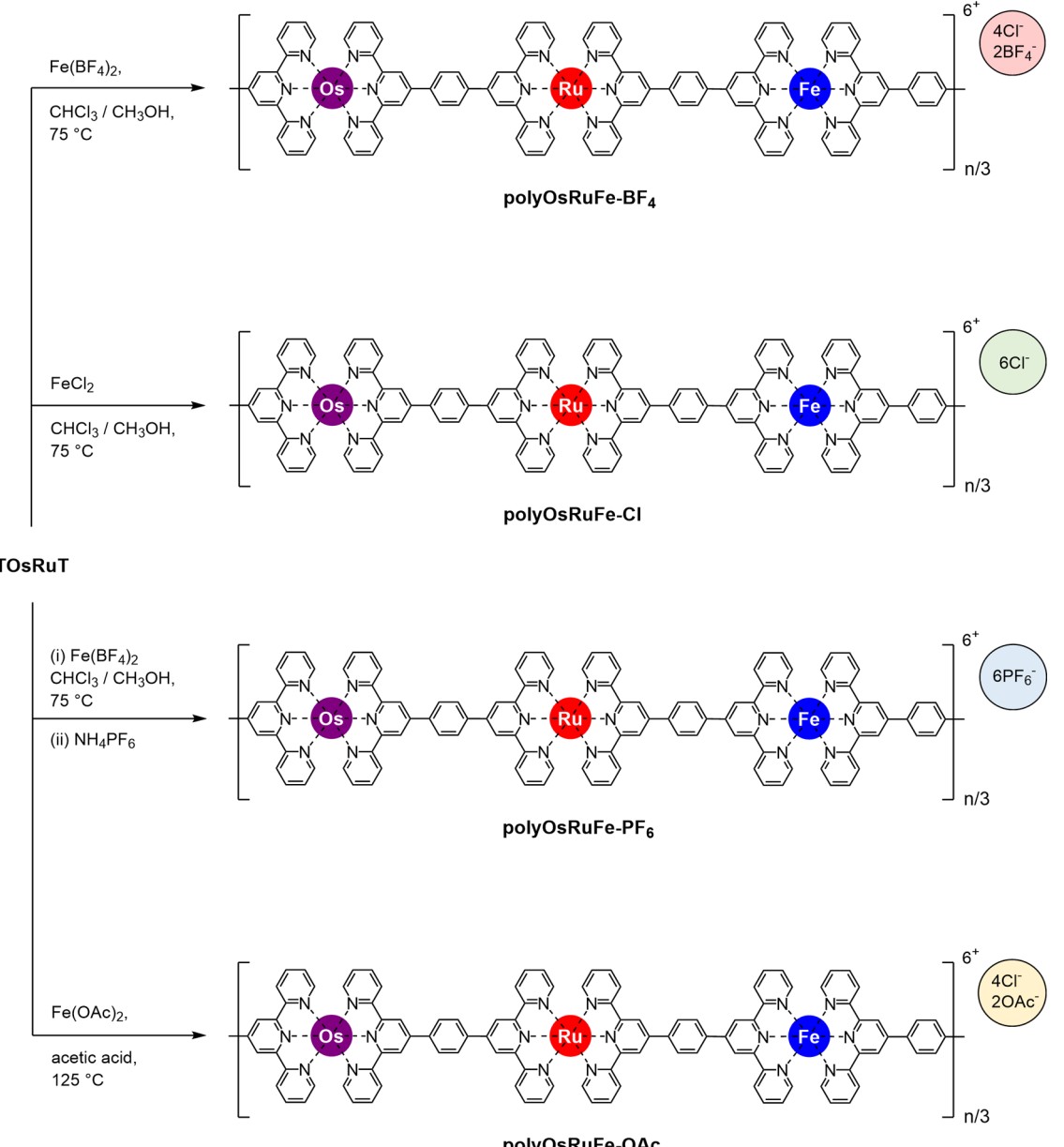

**Fig. 6 Synthesis and chemical structure of polyOsRuFe-A (A = BF₄, Cl, PF₆, and OAc).** Synthetic route to polyOsRuFe with various counteranions, called polyOsRuFe-A (A = BF₄, Cl, PF₆, and OAc).

The redox characteristics of polyOsRuFe-OAc were relatively unchanged, even after 1000 redox switching cycles, indicating its potential for various electrochemical and electro-optical applications. The solubility of the polyOsRuFe-As was tested in various solvents, as shown in Supplementary Table 6. A change in the counteranions from polyOsRuFe-BF₄ to polyOsRuFe-Cl did not affect the solubility of the polymer, but the counteranions affected the solubility of polyOsRuFe-PF₆ and polyOsRuFe-OAc. PolyOsRuFe-OAc was soluble in a low boiling solvent, $CH_3OH$, and in green solvents, such as EtOH and $H_2O$, which could broaden its applicability in diverse fields, including biological environments. Moreover, polyOsRuFe-PF₆ and polyOsRuFe-OAc showed different solubility; the former was soluble in $CH_3CN$ but insoluble in $CH_3OH$, EtOH, and $H_2O$, whereas the latter was soluble in $CH_3OH$, EtOH, and $H_2O$ but insoluble in $CH_3CN$ (Fig. 7). Hence, the solubility in various low boiling solvents could be varied by adjusting the counteranions of polyOsRuFe, making polymer processing easier for several applications, particularly for

making a thin film on a desired substrate, which is the prime requirement for various device-related applications[10,35,37].

**Preparation of thin film of polyOsRuFe-OAc on ITO/glass and its spectroelectrochemical study.** When heterometal ions were introduced into a linear MSP, the coupling of heterometallic segments could produce attractive features. For example, a heterobimetallic supramolecular polymer that displays tri-color electrochromism upon the stepwise oxidation of the metal ions was developed[10]. Electrochromism is defined as the optical color change of material upon a redox alteration, which has received tremendous interest in recent years for smart display applications[38–45]. MSPs generally display EC color changes upon redox changes to the metal center[4]. As polyOsRuFe (without considering the counteranions) showed a multiple and broad absorption window in the visible region and exhibited three distinct redox characteristics because of the presence of three redox-active metal ions [Fe(II), Ru(II), and Os(II)], its EC

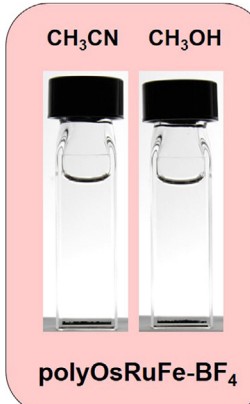
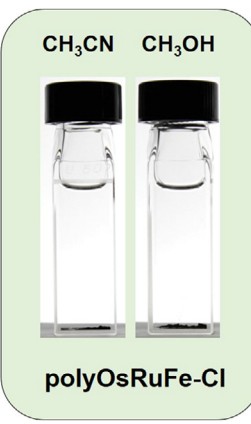
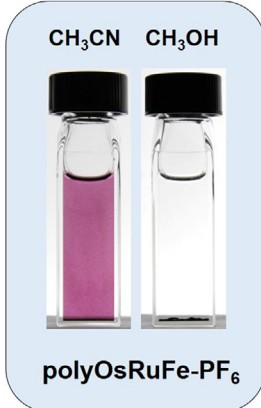
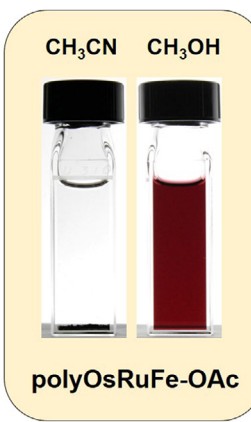

**Fig. 7 Observed solubility of polyOsRuFe-A (A: BF₄, Cl, PF₆, and OAc) in CH₃CN and CH₃OH.** The observed solubility of polyOsRuFe-BF$_4$, polyOsRuFe-Cl, polyOsRuFe-PF$_6$, and polyOsRuFe-OAc in CH$_3$CN and CH$_3$OH. Besides CH$_3$OH, the polyOsRuFe-OAc was also soluble in EtOH and H$_2$O.

behavior could be expected[10]. To validate this assumption, a spin-coating thin film of polyOsRuFe-OAc was prepared on ITO/glass (active area of the film: 1 cm × 1 cm) using its solution in CH$_3$OH. The spectroelectrochemical behavior of the film was investigated in a three-electrode system [polyOsRuFe-OAc/ITO/glass as the WE, platinum wire as the CE, and Ag/Ag$^+$ as the reference electrode (RE)] by monitoring the in situ UV–vis spectral change upon the application of a voltage (Fig. 8a, b and Supplementary Fig. 33a). Interestingly, the thin film of polyOsRuFe-OAc on ITO/glass exhibited quad-color electrochromism starting from magenta, to brown, to yellow, and to green upon the stepwise oxidation of Os(II) at 0.70 V, followed by Fe(II) at 0.85 V, and finally Ru(II) at 1.20 V with a gradual change in the MLCT absorption bands (Fig. 8c, d). Initially, the color of the polyOsRuFe-OAc film on ITO/glass was magenta due to the combination of MLCT absorption at 502 nm of <tpy-Os(II)-tpy> connectivity + <tpy-Ru(II)-tpy> connectivity and at 574 nm of <tpy-Fe(II)-tpy> connectivity (details of MLCT absorption peak has been discussed in an earlier section of Fig. 4c). When the oxidation of Os(II) occurs at 0.70 V, the MLCT bands for <tpy-Os(II)-tpy> connectivity disappeared, and the color of the film changed to brown due to the combined MLCT at 499 and 573 nm for <tpy-Ru(II)-tpy> connectivity and <tpy-Fe(II)-tpy> connectivity, respectively. After the oxidation of Fe(II) at 0.85 V, the MLCT band for <tpy-Fe(II)-tpy> connectivity disappeared, and the color of the film changed to yellow due to MLCT at 499 nm for <tpy-Ru(II)-tpy> connectivity. Finally, after the oxidation of Ru(II) at 1.20 V, the color of the film changed to green due to the disappearance of MLCT absorption of <tpy-Ru(II)-tpy> connectivity and the appearance of a new absorption at 401 nm (Supplementary Fig. 33b). In this context, after the first oxidation of Os(II), the absorption peak at 401 nm appeared, and its intensity increased gradually upon the stepwise oxidation of the remaining metal centers. Therefore, the 401 nm peak partially contributed to the color of the film after the first oxidation. This 401 nm peak may be due to the terpyridine-oxidized metal part, which increases gradually upon the successive oxidation of heterometal ions. The color of the film also changed reversibly to its initial state by the stepwise reduction of metal ions with the reappearance of the MLCT absorption bands. The EC switching stability of the polyOsRuFe-OAc film on ITO/glass was examined further for at least 300 cycles by applying a double-potential step (0.0 and 1.2 V) as a function of time (chronoamperometry, interval time: 5 s) and monitoring in situ the change in absorbance at 502 nm. Supplementary Fig. 33c and d shows the changes in absorbance of the polymer film. The absorbance of the film remained unchanged after 300 cycles, suggesting stable

EC color switching for many cycles. This quad-color electrochromism in an MSP has been realized for the first time. Thus far, only two heterometal ions were introduced into an MSP chain and displayed only tri-color electrochromism[10]. However, this HTMSP (polyOsRuFe) with three redox-active metal ions displayed quad-color electrochromism, which will certainly accelerate the development of more advanced voltage-tunable multicolor EC displays with unique features.

**Conclusions**. Three different transition metal ions were introduced to an MSP in a stepwise manner, and quad-color electrochromism was achieved. The heterometal ions were introduced into the polymer in homoleptic coordination environments made by two 2,2′:6′,2″-terpyridine (tpy) units. First, a strong coordination metal ion Os(II) complex was prepared, followed by the introduction of another strong coordination metal ion, Ru(II), to the Os(II) complex. Finally, a weak coordination metal ion, Fe(II), was bound to the Os(II)–Ru(II) complex to produce a linear MSP with the Fe(II)/Ru(II)/Os(II) sequence (polyOsRuFe). The weak coordination metal ion was varied to Co(II) and Zn(II) for the synthesis of HTMSPs with the Co(II)/Ru(II)/Os(II) and Zn(II)/Ru(II)/Os(II) sequence. The HTMSPs displayed a broad optical and electrochemical window because of a combination of three heterometallic segments into an MSP chain, which is expected to broaden their potentiality to various electro-optical applications. The solubility of polyOsRuFe in various low boiling solvents, including green solvents, could be tuned by adjusting the counteranions, which will make the processability of polymer easier for several applications. Moreover, the spin-coated thin film of polyOsRuFe on ITO/glass exhibited quad-color reversible EC changes upon the stepwise oxidation/reduction of Os(II), Fe(II), and Ru(II). This unique quad-color electrochromism of polyOsRuFe could be used to construct single MSP-based voltage-tunable multicolor EC devices, which is currently under development in the authors' lab.

Overall, this paper reported a strategy to integrate three homoleptic heterometal complexes into a linear MSP chain to merge their optical and electronic properties and produce functional materials. In addition, the successful integration of Os(II) and Ru(II) complexes with other diverse metal ions was realized, which could broaden the application of these materials to areas, such as artificial photosynthesis and light-harvesting antennae, because of the strong light absorption in the visible spectrum and long-lived MLCT excitation of Os(II) and Ru(II) metal centers[46–48]. Considering the huge scope in the molecular design of the ligand and the availability of various metal ions, this work will pave the way to producing heterometallic

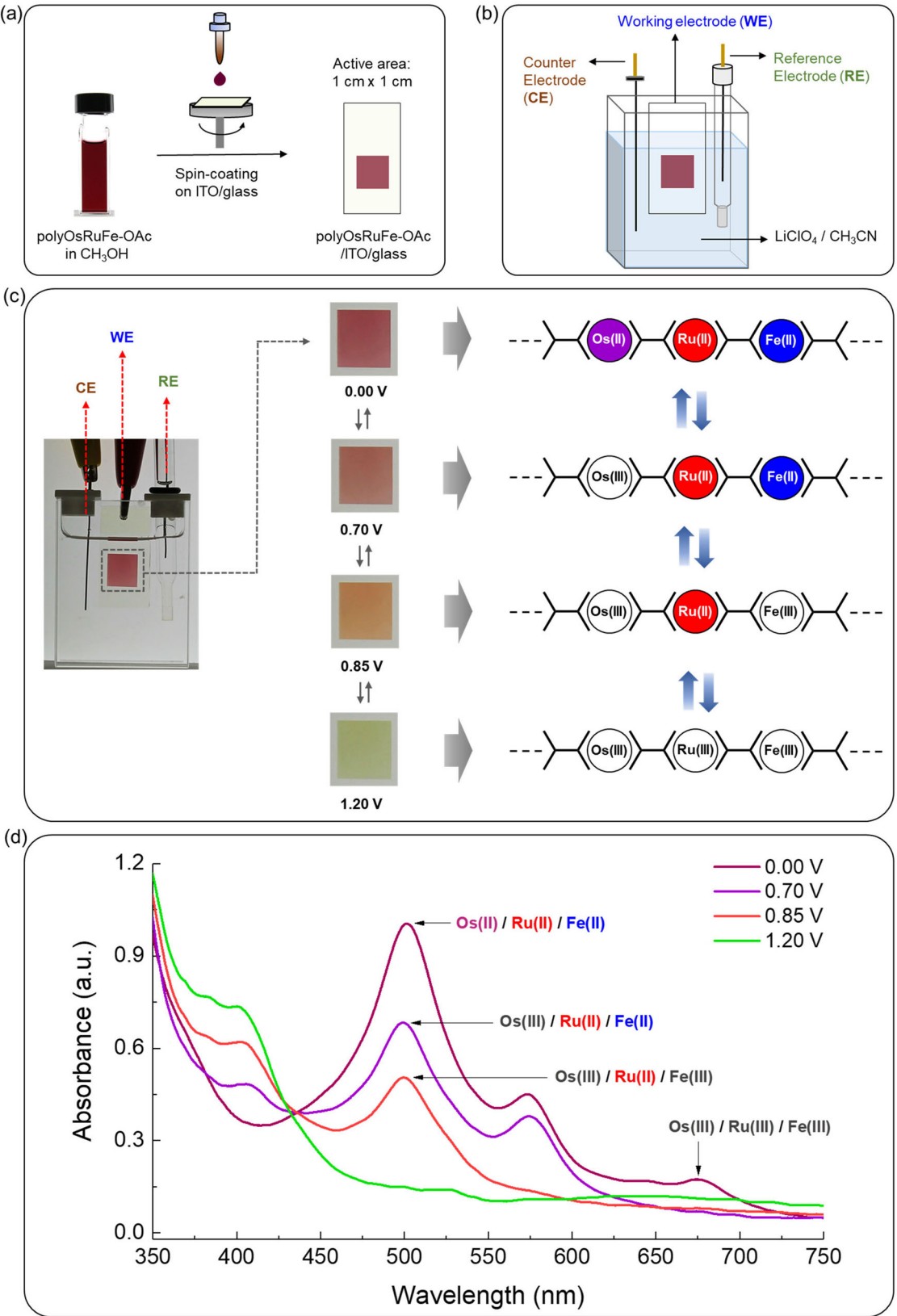

**Fig. 8 Preparation of a thin film of polyOsRuFe-OAc on ITO/glass and in situ spectroelectrochemical measurements. a** Schematic view of the preparation of spin-coating thin film on ITO/glass using $CH_3OH$ solution of polyOsRuFe-OAc. **b** Schematic representation of the three-electrode electrochemical measurement setup; polyOsRuFe-OAc/ITO/glass as WE, a platinum wire as the CE, and Ag/Ag$^+$ as the reference electrode (RE). **c** Actual view of the three-electrode electrochemical cell with a side panel showing photographs of the film (1 cm × 1 cm) at different applied potentials. Side bar: the mechanistic view of the polymer for the color change is shown schematically. **d** In situ UV–vis spectra of a polyOsRuFe-OAc film on ITO/glass at different applied potentials of 0.70, 0.85, and 1.20 V for stepwise oxidation of Os(II), Fe(II), and Ru(II), respectively.

supramolecular polymers with more structural diversity and complexity to discover more functionalities.

## Methods

**Materials**. All reagents and solvents were purchased from commercial sources (Aldrich Chemical Co., TCI Co., Wako, and Kanto Chemical Co., Inc.) and used as received. Column chromatography was performed using $SiO_2$ 60 (100–200 μm) from Kanto Chemical Co. Inc. and basic $Al_2O_3$ [Brockman Activity I (60–325 mesh)] from Wako Chemical. Anhydrous grade solvents were used for synthesis, and spectrophotometric grade solvents were used for the spectroscopic measurements. Distilled water prepared using a Milli-Q purification system was used for the synthesis and experiments.

**Instrumentation**. The $^1H$, $^{13}C$, 2D COSY, and NOESY NMR experiments were performed on a JEOL-ECZ 400 MHz NMR instrument. The chemical shifts are shown relative to tetramethylsilane and are expressed in parts per million. The abbreviations for signal multiplicities are expressed as follows: s for singlet, d for doublet, t for triplet, m for multiplet, br for broad, and brm for broad multiplet. The mass spectra were recorded using a Shimadzu LCMS-IT-TOF spectrometer. The UV–vis spectra were recorded on a Shimadzu UV-2550 UV-visible spectrophotometer. XPS (PHI Quantera SXM, (ULVAC-PHI) was performed using monochromatic Al Kα X-rays ($1.4 \times 0.1$ mm 100 W (20 kV, 5 mA), take-off angle of 45°, survey spectra pass energy of 280 eV, and energy step of 0.5 eV). The binding energies were calibrated to the C1s peak at 285.0 eV. The molecular weight of the polymers was determined via RALLS on a Viscotek 270 Dual Detector instrument in DMSO (flow rate: 0.50 mL/min). TGA (SII TG/DTA 6200) of the polymers was conducted in an $N_2$ environment with a heating rate of 10 °C/min. FTIR spectroscopy (Nicolet 4700 Ftir) was conducted using a mercury–cadmium telluride detector, and the transmittance measurement was monitored using a KBr disk. Cyclic voltammetry (CV) was performed using an ALS/CHI electrochemical workstation (CH Instruments, Inc.). A conventional three-electrode system (sample drop cast on freshly polished glassy carbon as the WE, platinum wire as the CE, and Ag/Ag$^+$ electrode in acetonitrile with 0.1 M TBAP + 0.01 M AgNO$_3$ as the reference electrode) was used for the CV study. A 0.1 M lithium perchlorate (LiClO$_4$) solution in CH$_3$CN was used as the electrolyte solution for all intermediate compounds and HTMSPs except for polyOsRuCo and polyOsRuFe-PF$_6$, for which A 0.1 M tetrabutylammonium perchlorate (TBAP) in acetone was used as the electrolyte. An integrated Ocean Optics modular spectrometer connected to the electrochemical analyzer was used for the in situ UV–vis absorption measurement of a polyOsRuFe-OAc film on ITO/glass upon the stepwise alteration of the voltages (spectroelectrochemical measurement).

**Synthesis**. For the synthesis and characterization of all intermediate compounds and polymers, see Supplementary Methods.

## Data availability

All data supporting the findings of this study are available within the article (and Supplementary Information Files) or available from the corresponding author on reasonable request.

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

## Acknowledgements

The authors sincerely acknowledge Namiki Foundry Station and Materials Analysis Station, NIMS, Tsukuba, for providing the instrumental facility. The authors also wish to acknowledge the CREST project (grant number: JPMJCR1533) from the Japan Science and Technology Agency for financial support.

## Author contributions

M.K.B. designed the concept, performed the experimental synthesis and characterizations, and wrote the manuscript. Y.N. synthesized compound 2 and measured the molecular weight of the polymers. M.H. edited the manuscript. All authors discussed the results and commented on the manuscript.

## Competing interests

The authors declare no competing interests.
