## [Peer Review File · Communications Chemistry]

Reviewers' comments:

Reviewer #1 (Remarks to the Author):

Review attached.

Reviewer #2 (Remarks to the Author):

The manuscript entitled "Sequence Control of Three Different Transition Metals in Metallo-Supramolecular Polymer and the Quad-Color Electrochromism by M. Higuchi et al. reported on the synthesis and full characterization of a heteronuclear trimetallic coordination polymer, and on the investigation of its optical and electrochemical properties, also in the form of thin films.

The work is interesting, it goes along the auspicated direction, not yet fully explored, of building heterometallic coordination polymers, thus deserves publication. The synthetic strategy adopted, selected after failed attempts well described in the SI, is interesting, and the characterization of all of the intermediate species is very well done.

I have however a few concerns which should be commented by the authors:

1) Despite the success in making compound TOsRuT, the synthetic strategy towards the final coordination polymer leads to a mixture of species where, within the linear arrays, the sequence of metals is not unique (see Fig. 2 on page 11). This poses a question of whether the "precise synthesis of the polymer" could be claimed, as stated, and whether the strategy described can be in the future modified in order to fix this problem.

Also, the way the authors describe this problem should be made clearer (writing that "the polymer may be formed in two possible structures" is quite misleading).

2) As to Fig. 1 b, I was not expecting to see a plateau reached at ca. 1 equivalent of Fe(II) ions added, as the compound TOsRuT has two binding sites which are quite far apart and probably independent.

3) The last concern is related to the solubility of the final coordination polymers. As the arrays are linear and rigid, I was expecting to find these materials very insoluble in all solvents. I am surprised that it is instead found that arrays of ca. 2500-6000 units ($5.44 \times 10^6 - 1.35 \times 10^7$ Da, estimated by RALLS, depending on counteranions) are soluble.

Is there any other way to confirm the presence in solution of such high MW polymeric species?

On the same note, Fig. 3c shows a peak at 575 nm for the absorption of the polymeric species, which is however in line with what seen in Fig. 1, where I suspect no polymer is formed (no solubility in CH₃OH, see supplementary table 5).

As a minor comment, the readability of the manuscript could benefit from a check of the English language.

In summary, I would be able to recommend this manuscript for publication in Communications Chemistry only after the authors have carefully responded to the points raised above.

Reviewer #3 (Remarks to the Author):

The manuscript "Sequence Control of Three Different Transition Metals in Metallo-Supramolecular Polymer and the Quad-Color Electrochromism" by Higuchi et al. is the first study to create metallo-supramolecular polymers in homoleptic environment with three separate metal ions, allowing for quad-control electrochromism. There was some literature reports on multi-color electrochromic materials based on 3 different metal centers but often in a heteroleptic environment or incorporated as isolated molecular complexes, not as part of the Metallo-supramolecular polymers.

This is a high-quality manuscript that I found was very interesting to read. It deserves to be published and will be interesting reading for a broad audience.

To the extent of my knowledge, it is synthetically challenging to selectively incorporate different metal centers in a fully controlled fashion in the oligomeric/polymeric units. The authors present interesting and very detailed synthetic methodologies to do this. Multiple sequences and combinations of metal ions were achieved. An additional strength of the paper is the fact that in-depth synthetic discussion was not limited to only the successful synthetic steps for the obtainment of desired polymers, but additionally includes the unsuccessful synthetic routes with a discussion regarding why these pathways were not successful. The inclusion of this information is equally informative for the transfer of knowledge. I greatly enjoyed reading this [part of the manuscript]. Furthermore, the NMR characterization including 2D spectra and proper signal labelling is truly well done. Authors employ very detailed NMR studies to analyze their molecules and materials with excellent purity spectral data and professional analysis of the spectra. While the synthetic part of the manuscript is exceptionally well done, the electrochemical analysis is very brief. I would suggest accepting the manuscript after some minor revisions and improvements/ additional experiments suggested below.

The electrochemical analysis in this manuscript requires some improvement prior to publication. Supplementary tables 2 and 4 are likely mislabeled. All spectroelectrochemical data suggests that the order of oxidation is Os → Fe → Ru as the material is swept to higher potentials. This result aligns with the author's figure design. However, both tables are labelled such that oxidation occurs in the order of Os → Ru → Fe which is not aligned to the rest of the manuscript. Please correct.

The cyclic voltammograms included in the SI all obtain different current densities (sometimes double), where the peak height is related to the amount of electrochemically accessible material. For example, SI Fig. 21, 30, 31, and 32. The authors should briefly discuss the discrepancies with the current densities between the different CVs. The result might be due to these two factors: human error in the drop casting volume, or a result caused by the diffusion of the counter ions to and from the working electrode. Because the CVs in question involve different counter anions, the latter is a strong possibility. Additionally, the individual peak height ratios should be considered between metal to metal. For example, In SI Fig. 30a and 32a, there is an approximately 1:1 ratio between Os to Fe, but more current at the Ru oxidative potential. In SI Fig. 31 a, Os and Ru are approximately 1:1 in peak height, but Fe achieves lower current densities. This may be related to the surface coverage, which can be evaluated through a study of the CVs at different scan rates (ex. inclusion of more CV sweep rates in addition to 50 mV/s).

The authors have not included any redox stability considerations for their material. The manuscript would be improved using a CV durability test, which would support the electrochemical reversibility

of the main polymer (polyOsRuFe-OAc). For example, the durability cycling would reveal the robustness of this polymer, or on the contrary, this experiment may show whether certain metal centers would begin to degrade earlier than others. Even if degradation occurs during this long term cycling, it would give interesting information about the system. I would be very interested to see how many electrochemical cycles could survive each metal center in the material. Would the stability be truly dependent on the nature of the metal center?

The rationale behind the application of Os(II), Fe(II), Ru(II), is clear, as they experience coloured MLCT and d-d* transitions with terpyridine coordinating ligand. The authors should explain briefly the rationale behind the incorporation of Zn(II) into the electrochromic polymer.

This manuscript (COMMSCHEM-20-0418-T) by Higuchi and coworkers demonstrated an excellent synthetic approach to access well-engineered hetero-trimetallic assembly in solution. The synthetic work deserves special credit in this manuscript. The problem and aim of the work were explained nicely, and the importance of ready access of well-engineered redox-active multi-metallic assembly for developing future voltage-tenable multi-electrochromic materials and devices was explained clearly. The claims and conclusions are coherent. I agree fully with the fact that assembly of multiple, different, properly tunable redox-active metal complexes into a processable polymeric/any other composite material is highly important when looking toward development of voltage-tuned electrochromic and similar devices. This work certainly fits for this journal considering the novelty, importance, competence, and standard/quality of the research described herein. However, according to the opinion and assessment of this reviewer, some crucial points (noted below) need to be addressed in the revision cycle before acceptance. These suggestions are intended to not only improve the competence, quality and strength of the current version but also to benefit the other peer researchers in terms of reproducibility and validity of the work.

<1> Page 3, 2nd paragraph, Line 1: “We noticed that a reaction of a metal complex with an organic compound was useful to.” What does it mean? Is it incomplete somehow to convey the intended message? Please check.

<2> The authors stated that they accomplished “stepwise harness of strong coordination metal ion Os(II) followed by another strong coordination metal ion Ru(II) and then weak coordination metal ion Fe(II)”. It would be better to provide the typical binding constant values of $[\text{Os}(\text{tpy})_2]^{2+}$, $[\text{Ru}(\text{tpy})_2]^{2+}$, and $[\text{Fe}(\text{tpy})_2]^{2+}$ complexes which are known in literature, to have a quantitative idea of the above statement.

<3> Scheme 1: The reaction **TOsT** to **TOsRuBr** should show the time (15 h in this case), and also the molar ratio of **TOsT**: **complex 4** employed or this reaction. Why was the Ru(Br-tpy) unit not attached to both the tpy ends of the **TOsT** molecule, even if 1:1 molar ratio was used (I see from the supporting information)? I guess the reaction would not be so selective, and some bis-Ru complex would form via 1:2 reaction (molar ratio = 0.5**complex 4** + 1**TOsT** + 0.5**complex 4**). When you purify the **TOsRuBr** compound by chromatography, did you not observe and separate any such bis-Ru-species? If yes, please mention in the manuscript/supporting information so that future researchers will not be confused.

<4> The authors mentioned that “the **polyOsRuFe** was synthesized by complexation of **TOsRuT** with $\text{Fe}(\text{BF}_4)_2 \cdot 6\text{H}_2\text{O}$ (1:1 molar ratio of ligand and metal) in a mixture solvent of CHCl_3 and CH_3OH (1:1, v/v) at 75 °C for 24 h, which gave the final product as precipitate with 90 % yield. The **polyOsRuFe** was mainly soluble in high boiling solvent such as DMSO and DMF.” However, just before the synthesis, the UV-vis titration experiment of **TOsRuT** with $\text{Fe}(\text{BF}_4)_2$ was conducted at 25 °C in $\text{CH}_2\text{Cl}_2/\text{CH}_3\text{OH}$ mixed solvent. So, I should expect a precipitate if the polymer is formed as mentioned in the synthetic procedure! Was the product **polyOsRuFe** not precipitated, if it formed at all? In such case, how did they ensure the accuracy and reliability of this titration experiment to conclude that 1:1 molar ratio is needed for polymerization? If no precipitate was formed, why so? The authors should clarify this contradictory issue.

<5> Why was no mass spectrometric (ESI, or MALDI or TOF-SIMS) experiment attempted for **polyOsRuFe**?

<6> The CV (Fig. 3d) of the **polyOsRuFe** polymer shows that the three different redox active centres do not communicate. Is it so? Any comment? A statement in the relevant texts would be welcome. The solvent used in the CV study should be mentioned in the Figure caption. Was the scan rate varied in the CV studies? If not, it should have been checked.

<7> In the context of inter-metallic communication, I see that in the CV of **TOsRuT**, the $E_{1/2}$ value of Os(II)/Os(III) pair was 0.63 and of Ru(II)/Ru(III) pair was 0.99 V. Now for the **polyOsRuFe**, the CV showed the values of 0.69 and 1.01 V for Os(II)/Os(III) and Ru(II)/Ru(III) redox pair, respectively. Interestingly, the CV of **polyOsRuCo** showed the values of 0.58 and 0.91 V for Os(II)/Os(III) and Ru(II)/Ru(III) redox pair, respectively. So we see that both Os(II)/Os(III) and Ru(II)/Ru(III) redox potential values modulated based on the change of the third metal centre at the remote site (none in **TOsRuT**, Fe(II) in **polyOsRuFe**, and Co(II) in **polyOsRuCo**). This proved that there was some communication among the redox centres in the systems. Can the authors probe into the extent of communication? Especially, an insight into the fact that while the Ru(II)/(III) redox potential increased (from 0.99 V in **TOsRuT** to 1.01 V in **polyOsRuFe**) when Fe(II) was incorporated at the remote site but decreased (from 0.99 V in **TOsRuT** to 0.91 V in **polyOsRuCo**) when Co(II) was incorporated at the same site, would be appreciated by the readers.

<8> In this regard, I would also suggest the authors to compare the observed redox potential values as well as CV behavior with reported 'discrete' OsRuFe (or discrete Os-Ru, Ru-Fe, Os-Fe) supramolecules with similar polypyridyl ligands. The difference and similarity should be emphasized.

<9> In the electrochromic study, what was the new absorption peak at 401 nm, appeared upon oxidation of Os(II) in the **polyOsRuFe-OAc** film coated on ITO, due to? In this regard, after all subsequent oxidation of the other metals, why the intensity of this peak at 401 nm increased? The authors did not discuss the reason at all. Is this kind of behavior (appearance of this 401 peak) common and found in pure $[\text{Os}(\text{tpy})_2]^{2+}$ complex/film? The authors may check if it was due to any MMCT event? A brief discussion on this issue would be beneficial to the readers.

<10> The solution-based (3-electrode system) quad-color electrochromism study: Was the redox cycle of the spin-coated film on ITO run for just one time? Have you checked multiple runs of the same film? If yes, what was the observation, especially on the reversibility and stability? If not, it should be shown.

<11> The texts which cite the references 37-41, should include the following recent bis-terpy-Fe-based ECD references as well: ACS Appl. Mater. Interfaces 2020, 12, 35181; Chem. Commun. 2020, 56, 559; Adv. Electron. Mater. 2020, 6, 2000407.

<12> Supplementary Table 5: Is the solvent "CH₃Cl" correct? I think it is "CHCl₃".

<13> Supplementary File, Section 1.9: Please mention the solvent used in the spectro-electrochemistry experiment.

<14> Supplementary File: All ESI-MS spectra should have first the "full" spectrum followed by zoomed part of the relevant section of the cation. Also, the isotopic pattern, both experimental and calculated, should be shown for the relevant mass peak.

<15> In Supplementary Figures 12, 17 (and also Figure 5), the differences between the exp. and calcd. mass values are not within error range normally accepted for HR-MS. For LR-MS, these are fine. Is there any explanation for the same?

-----*end of report*-----

Answer for Reviewer-1's Comments

Remark: *This manuscript (COMMSCHEM-20-0418-T) by Higuchi and coworkers demonstrated an excellent synthetic approach to access well-engineered hetero-trimetallic assembly in solution. The synthetic work deserves special credit in this manuscript. The problem and aim of the work were explained nicely, and the importance of ready access of well-engineered redox-active multi-metallic assembly for developing future voltage-tenable multi-electrochromic materials and devices was explained clearly. The claims and conclusions are coherent. I agree fully with the fact that assembly of multiple, different, properly tunable redox-active metal complexes into a processable polymeric/any other composite material is highly important when looking toward development of voltage-tuned electrochromic and similar devices. This work certainly fits for this journal considering the novelty, importance, competence, and standard/quality of the research described herein. However, according to the opinion and assessment of this reviewer, some crucial points (noted below) need to be addressed in the revision cycle before acceptance. These suggestions are intended to not only improve the competence, quality and strength of the current version but also to benefit the other peer researchers in terms of reproducibility and validity of the work.*

Reply: We are highly thankful to the reviewer-1 for reviewing and spending your valuable time with our manuscript. We are also grateful for your excellent remark and positive assessment about our manuscript. Thank you for your valuable suggestions to revise our manuscript. We have revised our manuscript accordingly.

Comment 1: *Page 3, 2nd paragraph, Line 1: "We noticed that a reaction of a metal complex with an organic compound was useful to." What does it mean? Is it incomplete somehow to convey the intended message? Please check.*

Reply: Many thanks for your pointing out. We corrected the sentence as follows.

In Page 3, "*We noticed that a reaction of a metal complex with an organic compound was useful to create a metal-containing ligand which can undergo further complexation with heterometals ions to give heterometallic supramolecular complexes or polymers. Therefore, we assumed that similar strategy could be utilized to organize three heterometal ions into MSP.*"

Comment 2: *The authors stated that they accomplished "stepwise harness of strong coordination metal ion Os(II) followed by another strong coordination metal ion Ru(II) and then weak coordination metal ion Fe(II)". It would be better to provide the typical binding constant values of [Os(tpy)₂]²⁺, [Ru(tpy)₂]²⁺, and [Fe(tpy)₂]²⁺ complexes which are known in literature, to have a quantitative idea of the above statement.*

Reply: Many thanks for your nice suggestion. According to your suggestion, we tried to find the binding constant value for [Os(tpy)₂]²⁺, [Ru(tpy)₂]²⁺, and [Fe(tpy)₂]²⁺ complexes. Unfortunately, we don't find any reports showing the exact value of the binding constant of three metal complexes (only for

[Fe(tpy)₂]²⁺ was found). However, we found two literatures (J. Mass Spectrom. 2003, 38, 510–516 and Eur. J. Inorg. Chem. 2015, 2015, 5662–5668) that clearly show the binding strength of [Os(tpy)₂]²⁺, [Ru(tpy)₂]²⁺, and [Fe(tpy)₂]²⁺ complexes follows the order Ru>Os>Fe. That mean Os(II) and Ru(II) form strong coordination compared with Fe(II). Here, we should mention one thing that although the study by Newkome et al. (Eur. J. Inorg. Chem. 2015, 2015, 5662–5668) revealed that the binding strength of [Os(tpy)₂]²⁺, [Ru(tpy)₂]²⁺, and [Fe(tpy)₂]²⁺ complexes follows the order Ru>Os>Fe, our synthetic strategy is not exactly based on the consideration of the order of the binding strength of metal ions. We have designed and developed the synthetic route by considering two things; reaction conditions of Os(II), Ru(II), and Fe(II) with tpy and the stability of tpy-M(II)-tpy connectivity. That is why, we first made complexation with Os(II), followed by Ru(II) and then Fe(II).

We have updated the text in revised manuscript by informing the reported result of these two reported literatures for giving a quantitative idea to the readers. We have modified this in introduction section as well as in result section as follows.

In Introduction (page 4), “*Here strong coordination refers to the binding strength of tpy-M(II)-tpy connectivity (where M=Ru or Os or Fe). A study by Schubert et al. has shown that the binding strength of tpy-Ru(II)-tpy connectivity is more than tpy-Fe(II)-tpy connectivity.³² Another study by Newkome et al. has demonstrated that the binding strength of tpy-Os(II)-tpy and tpy-Ru(II)-tpy connectivity is more compared with tpy-Fe(II)-tpy connectivity.³³ These studies confirm that Os(II) and Ru(II) form strong coordination with tpy compared with Fe(II).*”

In Results (page 7), “*The study by Newkome et al. demonstrated that the binding strength of tpy-M(II)-tpy connectivity (M=Ru or Os or Fe) follows the order Ru > Os > Fe.³³ This order of binding strength of the metal complex indicated that TOsRuT could be synthesized in two ways; either first complexation of Os(II) followed by complexation of Ru(II) or first complexation of Ru(II) followed by complexation of Os(II) as both the metal ions form complex at high temperature compared with Fe(II). But, both approaches as shown in Supplementary Fig. 2 were unsuccessful to get TOsRuT.*”

Comment 3: Scheme 1: The reaction TOsT to TOsRuBr should show the time (15 h in this case), and also the molar ratio of TOsT: complex 4 employed or this reaction. Why was the Ru(Br-tpy) unit not attached to both the tpy ends of the TOsT molecule, even if 1:1 molar ratio was used (I see from the supporting information)? I guess the reaction would not be so selective, and some bis-Ru complex would form via 1:2 reaction (molar ratio = 0.5complex 4 + 1TOsT + 0.5complex 4). When you purify the TOsRuBr compound by chromatography, did you not observe and separate any such bis-Ruspecies? If yes, please mention in the manuscript/supporting information so that future researchers will not be confused.

Reply: According to your suggestion, we have modified Scheme 1. Actually, the Ru(Br-tpy) can be attached in both side of TOsT. But we can get the desired product with high yield (one side coupling) by controlling the molar ration of the reactants to 1:1. Also, during column chromatography we collected the first fraction as the desired product. Following the first fraction, a second fraction of

undesired product was appeared, which was probably the product of both side coupling. We have updated this information in the revised manuscript as well as in revised supporting information (SI) as follows.

In Page 6, “*However, during separation of TOsRuBr by column chromatography, the first fraction was collected as the desired product (following first fraction, a second fraction of undesired product was appeared, which was probably the product of both side attachment of compound 4 with TOsT).*”

In Page 13 of Supporting Information, “*to collect first fraction as desired product (a second fraction of undesired product was appeared, which was probably the product of both side attachment of compound 4 with TOsT)*”

Comment 4: *The authors mentioned that “the polyOsRuFe was synthesized by complexation of TOsRuT with Fe(BF₄)₂·6H₂O (1:1 molar ratio of ligand and metal) in a mixture solvent of CHCl₃ and CH₃OH (1:1, v/v) at 75 °C for 24 h, which gave the final product as precipitate with 90 % yield. The polyOsRuFe was mainly soluble in high boiling solvent such as DMSO and DMF.” However, just before the synthesis, the UV-vis titration experiment of TOsRuT with Fe(BF₄)₂ was conducted at 25 °C in CH₂Cl₂/CH₃OH mixed solvent. So, I should expect a precipitate if the polymer is formed as mentioned in the synthetic procedure! Was the product polyOsRuFe not precipitated, if it formed at all? In such case, how did they ensure the accuracy and reliability of this titration experiment to conclude that 1:1 molar ratio is needed for polymerization? If no precipitate was formed, why so? The authors should clarify this contradictory issue.*

Reply: Many thanks for your concern about this matter. During study of complexation behavior of TOsRuT, we successively added Fe(BF₄)₂ to the solution of TOsRuT and recorded absorption spectra of the solution. Upon gradual addition of Fe(BF₄)₂, the solution of TOsRuT got saturated and the polymer was precipitated. We recorded the absorption spectra of the solution mixture until it forms the precipitate. The last point (and the absorption spectrum) shown in Fig. 1b and 1c was recorded before the precipitation of the solution mixture. From the titration and plot of [Fe(BF₄)₂]/[TOsRuT] ratio, we estimated the molar ratio to 1:1. We have updated the information about the precipitation of the polymer in our revised manuscript as follows.

In Page 10, “*(when the solution mixture was reached to saturation, the polymer was precipitated)*”

Comment 5: *Why was no mass spectrometric (ESI, or MALDI or TOF-SIMS) experiment attempted for polyOsRuFe?*

Reply: Thank you for your suggestion. Actually, we attempted MALDI experiment for polyOsRuFe previously (after following few recent literatures), but no fruitful result was obtained. That is why we measured molecular weight by RALLS. Anyway, still our effort is in progress. If we get success, the result will be explored in a later manuscript.

Comment 6: The CV (Fig. 3d) of the polyOsRuFe polymer shows that the three different redox active centres do not communicate. Is it so? Any comment? A statement in the relevant texts would be welcome. The solvent used in the CV study should be mentioned in the Figure caption. Was the scan rate varied in the CV studies? If not, it should have been checked.

Reply: We think the redox centers do not communicate here. We have mentioned the solvent including electrolyte in the Figure caption of our revised manuscript. We have not studied CV earlier with different scan rate. Now, according to your suggestion, we have done it. The results are shown in the below figures. The left CVs were updated as Fig. 3d of the revised manuscript and the right graph was added in SI as Supplementary Figure 21. b. The linear correlations between the peak current and the scan rate suggests surface-confined electrochemical redox process which is not restricted by slow electrolyte diffusion.

Figures. Cyclic voltammograms of polyOsRuFe-Cl with scan rates of 0.01-0.1 V/s in three-electrode system, electrolyte: 0.1 M LiClO₄ in CH₃CN, and linear correlations between the peak current and the scan rate during oxidation (top) and reduction (bottom) processes ($R^2 > 0.99$ for fitting).

And the following explanation was added in the revised manuscript.

In Page 15, “The scan-rate-dependent (0.01–0.1 V/s) CV study of polyOsRuFe displayed linear proportionality of the peak current with the scan rate (Fig. 3d and Supplementary Fig. 21b), suggesting a surface-confined electrochemical redox process which is not restricted by slow electrolyte diffusion.¹⁰”

Comment 7: In the context of inter-metallic communication, I see that in the CV of TOsRuT, the $E_{1/2}$ value of Os(II)/Os(III) pair was 0.63 and of Ru(II)/Ru(III) pair was 0.99 V. Now for the polyOsRuFe, the CV showed the values of 0.69 and 1.01 V for Os(II)/Os(III) and Ru(II)/Ru(III) redox pair, respectively. Interestingly, the CV of polyOsRuCo showed the values of 0.58 and 0.91 V for Os(II)/Os(III) and Ru(II)/Ru(III) redox pair, respectively. So we see that both Os(II)/Os(III) and Ru(II)/Ru(III) redox potential values modulated based on the change of the third metal centre at the remote site (none in TOsRuT, Fe(II) in polyOsRuFe, and Co(II) in polyOsRuCo). This proved that there was some

communication among the redox centres in the systems. Can the authors probe into the extent of communication? Especially, an insight into the fact that while the Ru(II)/(III) redox potential increased (from 0.99 V in TOSRuT to 1.01 V in polyOsRuFe) when Fe(II) was incorporated at the remote site but decreased (from 0.99 V in TOSRuT to 0.91 V in polyOsRuCo) when Co(II) was incorporated at the same site, would be appreciated by the readers.

Reply: Many thanks for your kind advice. Actually, we measured CV of TOSRuT to confirm the presence of Os(II) and Ru(II). The TOSRuT contains two free tpy units along with Os(II) and Ru(II) complex where as polyOsRuFe contains the complex of Os(II, Ru(II) and Fe(II). So, it seems difficult to compare the redox potential data between them. Still, we may think the less $E_{1/2}$ value of Os(II)/Os(III) and Ru(II)/Ru(III) pairs in TOSRuT compared with polyOsRuFe is due to the presence of free tpy units in TOSRuT.

In case of polyOsRuCo, the CV was measured using 0.1 M tetrabutylammonium perchlorate (TBAP) in acetone as electrolyte, whereas the CV of polyOsRuFe was measured in 0.1 M lithium perchlorate (LiClO₄) in CH₃CN as electrolyte. So, the redox potential of Os(II)/Os(III) and Ru(II)/Ru(III) pairs will vary in polyOsRuCo and polyOsRuFe as the electrolytes and solvents are different. So, no comparison between them can be made.

We tried to measure the electrochemical property of polyOsRuCo using 0.1 M lithium perchlorate (LiClO₄) in CH₃CN as electrolyte, but no redox property of Co was detected. This may be due to counter anion effect, which may suppress the oxidation/reduction of Co.

Comment 8: *In this regard, I would also suggest the authors to compare the observed redox potential values as well as CV behavior with reported 'discrete' OsRuFe (or discrete Os-Ru, Ru-Fe, Os-Fe) supramolecules with similar polypyridyl ligands. The difference and similarity should be emphasized.*

Reply: Many thanks for your kind advice. According to your advice, we have prepared the following table. There are very few examples we found in the literature. Still, we have tried and summarized the literatures that deal with complex and polymers with terpyridine ligand and combination on heterometallic segments. The results are added in our revised SI file (Supplementary Table 3). And the following explanation was added in the revised manuscript.

In Page 15, "*The observed redox potential of three heterometal ions in polyOsRuFe has been found to be comparable with the previously reported heterometallic supramolecular complexes and polymers (see Supplementary Table 3 for the details of comparison).*"

Supplementary Table 3. Comparison of the observed redox potential of Os(II), Fe(II) and Ru(II) in polyOsRuFe with previously reported heterometallic complexes and polymers containing identical coordinating ligand.

Supramolecular systems (discrete complexes/molecules and polymers) containing identical polypyridyl ligands and Os(II)Ru(II)Fe(II) / Os(II)Ru(II) / Os(II)Fe(II) / Ru(II)Fe(II) system	Cyclic voltammetry behavior (scan rate, electrolyte)	Redox potential ($E_{1/2}$; V) For Os(II)/Os(III)	Redox potential ($E_{1/2}$; V) For Fe(II)/Fe(III)	Redox potential ($E_{1/2}$; V) For Ru(II)/Ru(III)	Reference
Ru(II)-Fe(II) containing polymer	Two reversible redox waves (50 mV/s, 0.1 M LiClO ₄ in CH ₃ CN, vs. Ag/Ag ⁺)		0.77	0.93	Molecules 25 , 5261 (2020)
Os(II)-Fe(II) containing polymer	Two reversible redox waves (50 mV/s, 0.1 M LiClO ₄ in CH ₃ CN, vs. Ag/Ag ⁺)	0.55	0.72		Macromol. Rapid Commun. 41 , 1900384 (2020)
Ru(II)-Os(II) dinuclear complex	Two reversible redox waves (vs. SEC)	0.95		1.39	J. Chem. Soc., Faraday Trans. 92 , 2223-2238 (1996)
Ru(II)-Os(II) complex	Two reversible redox waves (50 mV/s, 0.1 M TBAP in DMF, vs. Fc/Fc ⁺)	0.56		0.90	Chem. Eur. J. 8 , 130-136 (2002)
Os(II)-Ru(II)-Fe(II) containing polymer (polyOsRuFe)	Three reversible redox waves (100 mV/s, 0.1 M LiClO ₄ in CH ₃ CN, vs. Ag/Ag ⁺)	0.58	0.76	0.92	This work

Comment 9: *In the electrochromic study, what was the new absorption peak at 401 nm, appeared upon oxidation of Os(II) in the polyOsRuFe-OAc film coated on ITO, due to? In this regard, after all subsequent oxidation of the other metals, why the intensity of this peak at 401 nm increased? The authors did not discuss the reason at all. Is this kind of behavior (appearance of this 401 peak) common and found in pure [Os(tpy)₂]²⁺ complex/film? The authors may check if it was due to any MMCT event? A brief discussion on this issue would be beneficial to the readers.*

Reply: Actually, this kind of peak is quite common for polypyridyl containing metallo-supramolecular polymers with Os and/or Ru and/or Fe. When the electrochromic property of polypyridyl containing supramolecular systems is studied on ITO, this type of peak appears after oxidation of metal center into the materials. This peak is mainly originated from the ligand-oxidized metal part. In our case, gradual oxidation of three different metal ion, the amount of ligand-oxidized metal part increases, which results gradual increase of the intensity of 401 nm peak. According to your suggestion, we have added this discussion in our revised manuscript as follows.

In Page 24-25, “**This 401 nm peak may be due to the terpyridine-oxidized metal part which gradually increases upon successive oxidation of the heterometal ions.**”

Comment 10: *The solution-based (3-electrode system) quad-color electrochromism study: Was the redox cycle of the spin-coated film on ITO run for just one time? Have you checked multiple runs of the*

same film? If yes, what was the observation, especially on the reversibility and stability? If not, it should be shown.

Reply: Thank you for your concern in this matter. Actually, we studied few more redox cycles of the spin coated polymer film on ITO. But we wanted to show the result in a later manuscript where we want to disclose details electrochromic property of the polymer and its device property. Anyway, according to your suggestion, here we have given the date for cyclic stability for at least 300 cycles. The result is shown below. They were added in SI as **Supplementary Figure 33. c & d**. If you see the redox stability of the polymer (polyOsRuFe-OAc), it is stable for at least 1000 switching cycles (see Supplementary Figure 32d). So, we expect, our material will show high durability in terms of redox/electrochromic switching, which will be explored later. And the following explanation was added in the revised manuscript.

In Page 25, “**We further studied the electrochromic switching stability of the polyOsRuFe-OAc film on ITO/glass for at least 300 cycles by applying a double-potential step (0.0 and 1.2 V) as a function of time (chronoamperometry, interval time: 5 s) and in situ monitoring the absorbance change at 502 nm. The absorbance change of the polymer film is shown in Supplementary Fig. 33c & 33d. The result indicated that the absorbance of the film remains unchanged after 300 cycles, suggesting a stable electrochromic color switching for many cycles.**”

Figures. (a) Change in absorbance of polyOsRuFe-OAc film on ITO/glass for few cycles, and (b) up to 300 cycles; monitored at 502 nm upon switching the potential between 0 and 1.2 V during chronoamperometry measurement.

Comment 11: The texts which cite the references 37-41, should include the following recent bis-terpy-Febased ECD references as well: ACS Appl. Mater. Interfaces 2020, 12, 35181; Chem. Commun. 2020, 56, 559; Adv. Electron. Mater. 2020, 6, 2000407.

Reply: Thank you for your kind suggestion. We have included these references in our revised manuscript. Please see **the references of 43, 44, and 45** in our revised manuscript.

Comment 12: *Supplementary Table 5: Is the solvent “CH3Cl” correct? I think it is “CHCl3”.*

Reply: We are sorry for this mistake. We have corrected it in the revised SI file (the table 5 have been changed to Table 6 for revision purpose).

Comment 13: *Supplementary File, Section 1.9: Please mention the solvent used in the spectroelectrochemistry experiment.*

Reply: Thank you for your suggestion. We have mentioned this in the revised SI (Page 50).

Comment 14: *Supplementary File: All ESI-MS spectra should have first the “full” spectrum followed by zoomed part of the relevant section of the cation. Also, the isotopic pattern, both experimental and calculated, should be shown for the relevant mass peak.*

Reply: Thank you for your kind suggestion. We have modified all ESI-MS spectra according to your valuable suggestion in our revised manuscript. Please see the revised manuscript. The results are shown below. They were included in the revised SI (**Supplementary Figures 5, 12 and 17**).

Figure A. ESI mass of TOsT.

Figure B. ESI mass of TOsRuBr.

Figure C. ESI mass of TOsRuT.

Comment 15: In Supplementary Figures 12, 17 (and also Figure 5), the differences between the exp. And calcd. mass values are not within error range normally accepted for HR-MS. For LR-MS, these are fine. Is there any explanation for the same?

Reply: This is probably because we calculated the mass considering the peak with the highest intensity. But now we modified it by the comparison with the theoretical peak pattern. Hope these results will be satisfactory. Thank you for your kind suggestion. Please check the revised figures (Figure A, B & C shown above) in SI.

Finally, thank you very much once again for taking precious time from your work and helping us in improving this manuscript. We have tried our best to address every single concern that you have raised. We sincerely hope that current fully revised manuscript would meet all your concerns.

Answer for Reviewer-2's Comments

Remark: *The manuscript entitled "Sequence Control of Three Different Transition Metals in Metallo-Supramolecular Polymer and the Quad-Color Electrochromism by M. Higuchi et al. reported on the synthesis and full characterization of a heteronuclear trimetallic coordination polymer, and on the investigation of its optical and electrochemical properties, also in the form of thin films.*

The work is interesting, it goes along the auspicated direction, not yet fully explored, of building heterometallic coordination polymers, thus deserves publication. The synthetic strategy adopted, selected after failed attempts well described in the SI, is interesting, and the characterization of all of the intermediate species is very well done.

I have however a few concerns which should be commented by the authors:

Reply: We are highly thankful to the reviewer-2 for reviewing and spending your valuable time with our manuscript. We are also grateful for your positive consideration of our manuscript for publication. We have considered the concerns raised by you and revised our manuscript according to your valuable suggestions.

Comment 1: *Despite the success in making compound TOsRuT, the synthetic strategy towards the final coordination polymer leads to a mixture of species where, within the linear arrays, the sequence of metals is not unique (see Fig. 2 on page 11). This poses a question of whether the "precise synthesis of the polymer" could be claimed, as stated, and whether the strategy described can be in the future modified in order to fix this problem.*

Also, the way the authors describe this problem should be made clearer (writing that "the polymer may be formed in two possible structures" is quite misleading).

Reply: The TOsRuT can be considered as a modified ditopic ligand containing two heterometal ions. Now, if TOsRuT undergoes to coordination-driven self-assembly with a third heterometal ions, the resultant polymer definitely will be a heterotrimetallic supramolecular polymer. We think, the synthesis could be claimed to precise synthesis as no other mode of synthesis is possible to introduce (as well as organize) three heterometal ions in a metallo-supramolecular polymer chain in homoleptic environment. Now, we extended our thought in a micro-scale observation of the structure of TOsRuT. We can consider the final ligand as asymmetric structure due to presence of one Os and one Ru center. Therefore, we anticipated that the polymer may be formed in either head-to-tail structure or a mixture of head-to-tail and head-to-head/tail to tail. However, if there is any head-to-head/tail-to-tail possibility, we may observe additional characteristic in UV/CV..etc. But we don't observe that kind of signature to proof our thought. But theoretically, it should be possible. That is why we mentioned the fact in a paragraph, so that the readers/researchers can get a clear idea about the possibilities (and to find/investigate more). We think, using bulk synthesis, this defect (if there?) can not be controlled. The only way, if anyone find any experimental evidence to support the above assumption.

We have modified the paragraph for Fig. 2 in our revised manuscript to make it more clear to the readers as follows.

In Page 13, “*It should be noted that if we look the chemical structure of the modified ditopic ligand TOSRuT closely, an asymmetric structure can be considered for this ligand due to presence of one Os(II) and one Ru(II) complex into the modified ditopic ligand. Therefore, the reactivity of two tpy units at the two ends of TOSRuT may differ little bit due to presence of two heterometal ions. In other way, the TOSRuT can be imagined as a structure with one side as head and another side as tail. Thus, when TOSRuT reacts with Fe(II) to make the polymer, it could be anticipated that the resultant polymer (polyOsRuFe) may be formed in regular head-to-tail structure (Fig. 2a) or regular structure with some irregular head-to-head/tail-to-tail structure (Fig. 2b). That means the sequence of Fe(II)/Ru(II)/Os(II) in polyOsRuFe may be repeated in alternate fashion or the sequence may be repeated with small irregularity. However, at present we did not find any experimental evidence to support this kind of assumption (theoretical prediction). More investigation in this direction is currently underway.*”

Comment 2: *As to Fig. 1 b, I was not expecting to see a plateau reached at ca. 1 equivalent of Fe(II) ions added, as the compound TOSRuT has two binding sites which are quite far apart and probably independent.*

Reply: Thank you for your concern about this matter. The plateau reached to not exactly 1 but it close to 1. Actually, for metallo-supramolecular polymers, when ditopic ligand is used for titration experiment (study of complexation behavior) with divalent metal ion, such kind of plateau appears (close to equivalent point). So, we think, in our case the same thing happened. As Fe(II) reacts at room temperature, so probability of binding of Fe(II) at two sides of TOSRuT is more to build a linear polymeric structure. Another thing is that as the both Os and Ru forms nonlabile complexes, the reactivity of terpyridines at two ends of TOSRuT will not differ so much we think.

During study of complexation behavior of TOSRuT, we successively added Fe(BF₄)₂ to the solution of TOSRuT and recorded absorption spectra of the solution. Upon gradual addition of Fe(BF₄)₂, the solution of TOSRuT got saturated and the polymer was precipitated. We recorded the absorption spectra of the solution mixture until it forms the precipitate. The last point (and the absorption spectrum) shown in Fig. 1b and 1c was recorded before the precipitation of the solution mixture. From the titration and plot of [Fe(BF₄)₂]/[TOSRuT] ratio, we estimated the molar ratio of the reactants is close to 1.

Comment 3: *The last concern is related to the solubility of the final coordination polymers. As the arrays are linear and rigid, I was expecting to find these materials very insoluble in all solvents. I am surprised that it is instead found that arrays of ca. 2500-6000 units (5.44×10^6 – 1.35×10^7 Da, estimated by RALLS, depending on counteranions) are soluble.*

Is there any other way to confirm the presence in solution of such high MW polymeric species?

On the same note, Fig. 3c shows a peak at 575 nm for the absorption of the polymeric species, which is however in line with what seen in Fig. 1, where I suspect no polymer is formed (no solubility in CH₃OH, see supplementary table 5).

As a minor comment, the readability of the manuscript could benefit from a check of the English language.

In summary, I would be able to recommend this manuscript for publication in Communications Chemistry only after the authors have carefully responded to the points raised above.

Reply: Yes, the final polymer is soluble in DMSO and DMF. However, by varying counteranions, we can tune its solubility in low boiling solvents for various application purpose.

Sometime MALDI mass can be employed. We also tried that, but no fruitful result was obtained. That is way, we determined by RALLS.

Actually, the polymer is not soluble in MeOH. During titration (Fig. 1), as we mentioned above, when the solution mixture reached to saturation, the polymer is precipitated (not soluble in MeOH or MeOH/DCM mixture).

Thank you for your suggestion. We have carefully checked the English language in our whole revised manuscript and SI.

Finally, thank you very much once again for taking precious time from your work and helping us in improving this manuscript. We have tried our best to address every single concern that you have raised. We sincerely hope that current fully revised manuscript would meet all your concerns.

Answer for Reviewer-3's Comments

Remark: *The manuscript "Sequence Control of Three Different Transition Metals in Metallo-Supramolecular Polymer and the Quad-Color Electrochromism" by Higuchi et al. is the first study to create metallo-supramolecular polymers in homoleptic environment with three separate metal ions, allowing for quad-control electrochromism. There was some literature reports on multi-color electrochromic materials based on 3 different metal centers but often in a heteroleptic environment or incorporated as isolated molecular complexes, not as part of the Metallo-supramolecular polymers.*

This is a high-quality manuscript that I found was very interesting to read. It deserves to be published and will be interesting reading for a broad audience.

To the extent of my knowledge, it is synthetically challenging to selectively incorporate different metal centers in a fully controlled fashion in the oligomeric/polymeric units. The authors present interesting and very detailed synthetic methodologies to do this. Multiple sequences and combinations of metal ions were achieved. An additional strength of the paper is the fact that in-depth synthetic discussion was not limited to only the successful synthetic steps for the obtainment of desired polymers, but additionally includes the unsuccessful synthetic routes with a discussion regarding why these pathways were not successful. The inclusion of this information is equally informative for the transfer of knowledge. I greatly enjoyed reading this part of the manuscript. Furthermore, the NMR characterization including 2D spectra and proper signal labelling is truly well done. Authors employ very detailed NMR studies to analyze their molecules and materials with excellent purity spectral data and professional analysis of the spectra. While the synthetic part of the manuscript is exceptionally well done, the electrochemical analysis is very brief. I would suggest accepting the manuscript after some minor revisions and improvements/ additional experiments suggested below.

Reply: We are very thankful to the reviewer-3 for reviewing and spending your valuable time with our manuscript. We are also grateful for your excellent remark about our manuscript. Thank you again for your recommendation to accept our manuscript after some minor revision and improvement. We have revised our manuscript according to your valuable suggestions. Please check the revised manuscript.

Comment 1: *The electrochemical analysis in this manuscript requires some improvement prior to publication. Supplementary tables 2 and 4 are likely mislabeled. All spectroelectrochemical data suggests that the order of oxidation is Os -> Fe -> Ru as the material is swept to higher potentials. This result aligns with the author's figure design. However, both tables are labelled such that oxidation occurs in the order of Os -> Ru -> Fe which is not aligned to the rest of the manuscript. Please correct.*

Reply: Thank you so much for bringing this point to our notice. We have corrected the tables in our revised manuscript. Please see the revised manuscript. Table 4 has been changed to Table 5 in our revised manuscript for revision purpose.

Comment 2: The cyclic voltammograms included in the SI all obtain different current densities (sometimes double), where the peak height is related to the amount of electrochemically accessible material. For example, SI Fig. 21, 30, 31, and 32. The authors should briefly discuss the discrepancies with the current densities between the different CVs. The result might be due to these two factors: human error in the drop casting volume, or a result caused by the diffusion of the counter ions to and from the working electrode. Because the CVs in question involve different counter anions, the latter is a strong possibility. Additionally, the individual peak height ratios should be considered between metal to metal. For example, In SI Fig. 30a and 32a, there is an approximately 1:1 ratio between Os to Fe, but more current at the Ru oxidative potential. In SI Fig. 31 a, Os and Ru are approximately 1:1 in peak height, but Fe achieves lower current densities. This may be related to the surface coverage, which can be evaluated through a study of the CVs at different scan rates (ex. inclusion of more CV sweep rates in addition to 50 mV/s).

Reply: Yes, you are right. This may be due to different thickness of the drop casted film as the polymers are soluble in different solvents. In addition, counteranions effect is also there.

According to your suggestion, we have studied scan rate dependent CV of each polymers for getting information about surface coverage. The results are shown below (Figs. A, B, C and D). They are included as the CVs of Fig. 3d in the revised manuscript and Supplementary Figure 21b, 30, 31, 32 in the revised SI. The result suggesting surface-confined electrochemical redox process which is not restricted by slow electrolyte diffusion.

And the following explanation was added in the revised manuscript. In Page 15, “The scan-rate-dependent (0.01–0.1 V/s) CV study of polyOsRuFe displayed linear proportionality of the peak current with the scan rate (Fig. 3d and Supplementary Fig. 21b), suggesting a surface-confined electrochemical redox process which is not restricted by slow electrolyte diffusion.”¹⁰

Figure A. Cyclic voltammograms of polyOsRuFe with scan rates of 0.01-0.1 V/s in three-electrode system, electrolyte: 0.1 M LiClO₄ in CH₃CN, and linear correlations between the peak current and the scan rate during oxidation (top) and reduction (bottom) processes (R² > 0.99 for fitting).

Figure B. Cyclic voltammograms of polyOsRuFe-Cl with scan rates of 0.01-0.1 V/s in three-electrode system, electrolyte: 0.1 M LiClO₄ in CH₃CN, and linear correlations between the peak current and the scan rate during oxidation (top) and reduction (bottom) processes ($R^2 > 0.99$ for fitting).

Figure C. Cyclic voltammograms of polyOsRuFe-PF₆ with scan rates of 0.01-0.1 V/s in three-electrode system, electrolyte: 0.1 M TBAP in acetone, and linear correlations between the peak current and the scan rate during oxidation (top) and reduction (bottom) processes ($R^2 > 0.99$ for fitting).

Figure D. Cyclic voltammograms of polyOsRuFe-OAc with scan rates of 0.01-0.1 V/s in three-electrode system, electrolyte: 0.1 M LiClO₄ in CH₃CN, and linear correlations between the peak current and the scan rate during oxidation (top) and reduction (bottom) processes ($R^2 > 0.99$ for fitting).

Comment 3: *The authors have not included any redox stability considerations for their material. The manuscript would be improved using a CV durability test, which would support the electrochemical reversibility of the main polymer (polyOsRuFe-OAc). For example, the durability cycling would reveal the robustness of this polymer, or on the contrary, this experiment may show whether certain metal centers would begin to degrade earlier than others. Even if degradation occurs during this long term cycling, it would give interesting information about the system. I would be very interested to see how many electrochemical cycles could survive each metal center in the material. Would the stability be truly dependent on the nature of the metal center?*

Reply: Thank you for your kind suggestion. According to your suggestion, we have measured the CV durability of main polymer (polyOsRuFe-OAc) for at least 1000 cycles. The result is shown below (Figure E) and included in revised SI as Supplementary Figure 32d. This result indicates that stability does not depend on metal centers and the polymer can be used for various electrochemical application due to its high redox stability. And the following explanation was added in the revised manuscript.

In Page 20-21, *“To check the redox stability of polyOsRuFe, we further examined the cyclic stability of polyOsRuFe-OAc (taking it as example, as this polymer was used for electrochromic application shown in later section) for 1000 cycles and the observed data is shown in Supplementary Figs. 32d. The result indicated that the redox characteristic of polyOsRuFe-OAc remains almost unchanged even after 1000 redox switching cycles, which indicating its potentiality for various electrochemical as well as electro-optical applications.”*

Figure E. CV durability of polyOsRuFe-OAc up to 1000 cycles with scan rates of 50 mV/s in three-electrode system, electrolyte: 0.1 M LiClO₄ in CH₃CN.

Comment 4: *The rationale behind the application of Os(II), Fe(II), Ru(II), is clear, as they experience coloured MLCT and d-d* transitions with terpyridine coordinating ligand. The authors should explain briefly the rationale behind the incorporation of Zn(II) into the electrochromic polymer.*

Reply: Actually, in this manuscript we wanted to show the successful synthesis of heterotrimetallic supramolecular polymer (polyOsRuFe) by developing a synthetic route to prepare a modified ditopic ligand (TOsRuT). To extend the family of heterotrimetallic supramolecular polymer and to show that TOsRuT can also bind other divalent metal ion, we extended the metal to Co(II) and Zn(II). That was our main aim.

In addition, we think this Zn (II) containing polymer (polyRuOsZn) may be interested in researchers who deal with emission property of Os/Ru/Zn system.

Finally, thank you very much once again for taking precious time from your work and helping us in improving this manuscript. We have tried our best to address every single concern that you have raised. We sincerely hope that current fully revised manuscript would meet all your concerns.

Reviewers' comments:

Reviewer #1 (Remarks to the Author):

This reviewer is fully satisfied with the revised version of the manuscript. This version can now be accepted for publication.

Reviewer #2 (Remarks to the Author):

This revised version of the manuscript entitled "Sequence Control of Three Different Transition Metals in Metallo-Supramolecular Polymer and the Quad-Color Electrochromism by M. Higuchi et al." is considerably improved, especially in the electrochemical part.

My only concern left is still related to the claim of having attained control over the sequence of metals within the polymeric structure. I suggest toning down this claim (see abstract, intro and conclusion) for in my opinion the authors do not present any definite proof of being able to do that. After this only modification is implemented, I would be glad to recommend the manuscript for publication in Communications Chemistry.

The reason for my comment above is briefly the following:

To my understanding, in the case that, as it is suggested, "the reactivity of two tpy units at the two ends of TOSRuT may differ little bit due to presence of two heterometal ions" (pag.13), it would be important to define that "little bit", otherwise the most probable situation is that of a complete random sequence, and thus no control over the assembly.

If the two sites act very differently instead (say the Ru side binds Fe a lot better), then a controlled sequence could be within reach (first step of the assembly process would be the formation of OsRuFeRuOs "dimers" which then would lead to an ordered (OsRuFeRuOs-Fe-OsRuFeRuOs)_n polymeric sequence.

Although the control of the sequence might be not so relevant for the final properties of the material (if the communication between neighboring centres is not very effective) it indeed remains an interesting point in terms of understanding the system and future designs. I am looking forward to reading more about developments on that in the authors' future investigations.

Reviewer #3 (Remarks to the Author):

The authors carefully addressed all comments raised by me and all other referees. I would suggest publishing this manuscript in its current version.

Our reply to Reviewer-1

Remark: *This reviewer is fully satisfied with the revised version of the manuscript. This version can now be accepted for publication.*

Our reply: Many thanks for your recommendation for publication.

Our reply to Reviewer-2

Remark: *This revised version of the manuscript entitled “Sequence Control of Three Different Transition Metals in Metallo-Supramolecular Polymer and the Quad-Color Electrochromism” by M. Higuchi et al. is considerably improved, especially in the electrochemical part.*

My only concern left is still related to the claim of having attained control over the sequence of metals within the polymeric structure. I suggest toning down this claim (see abstract, intro and conclusion) for in my opinion the authors do not present any definite proof of being able to do that.

After this only modification is implemented, I would be glad to recommend the manuscript for publication in Communications Chemistry.

The reason for my comment above is briefly the following:

To my understanding, in the case that, as it is suggested, “the reactivity of two tpy units at the two ends of TOsRuT may differ little bit due to presence of two heterometal ions” (pag.13), it would be important to define that “little bit”, otherwise the most probable situation is that of a complete random sequence, and thus no control over the assembly. If the two sites act very differently instead (say the Ru side binds Fe a lot better), then a controlled sequence could be within reach (first step of the assembly process would be the formation of OsRuFeRuOs “dimers” which then would lead to an ordered (OsRuFeRuOs-Fe-OsRuFeRuOs)_n polymeric sequence. Although the control of the sequence might be not so relevant for the final properties of the material (if the communication between neighboring centres is not very effective) it indeed remains an interesting point in terms of understanding the system and future designs. I am looking forward to reading more about developments on that in the authors' future investigations.

Reply: Many thanks for your recommendation for publication and your valuable suggestion about our claim of “sequence control”. According to your suggestion, we have toned down the claim of “sequence control” in Title, Abstract, Introduction, Main text (Page 13), and Conclusion as follows.

Title:

We changed “Sequence Control of Three Different Transition Metals in Metallo-Supramolecular Polymer and the Quad-Color Electrochromism” to “Stepwise Introduction of Three Different Transition Metals in Metallo-Supramolecular Polymer and the Quad-Color Electrochromism”.

Abstract:

We changed the words of “sequence control” in the 2nd and 3rd sentences of “Utilizing different complexation ability among transition metals enabled sequence control of three different transition metals in metallo-supramolecular polymer (MSP). A sequence-controlled MSP with Os, Ru, and Fe (polyOsRuFe) was synthesized as follows.” to “Utilizing different complexation ability among transition metals enabled stepwise introduction of three different transition metals in metallo-supramolecular polymer (MSP). An MSP with Os, Ru, and Fe (polyOsRuFe) was synthesized as follows.”. We also shorten the abstract within 200 words by deleting some sentences.

Introduction:

1) We deleted the following first 6 sentences about the explanation of sequence control in nature and polymer.

“Sequence control of monomers in polymer is important, especially in nature. For example, DNA encodes information through the sequence of the nucleotides along the polymer strand. Therefore, sequence control in polymer always receives much attention of scientists. However, the sequence control of monomers in the synthesis of polymers has been mainly investigated to the covalently bonded polymers. As for coordination polymers, however, the control of the metal sequence is not so easy, because the coordination process depends on the equilibrium in solution. The competition of complexation and decomplexation makes it difficult to fix the sequence of several metal ion species in the polymer.”

2) Instead of deleting the above sentences, we added the explanation of metallo-supramolecular polymers and our motivation to the research as follows.

“Metallo-supramolecular polymers (MSPs), which are synthesized by a 1:1 complexation of metal ion and ditopic ligand, have received attention due to a wide range of the applications including electrochromic displays, memory devices, sensors, energy storage devices, and anticancer therapies.¹⁻⁷ The electronic interaction between the metal and the ligand in the polymer chains causes unique electrochemical, optical, emissive properties unlike the conventional organic polymers. In general, one metal species is included in MSPs, but introduction of two metal ion

species in the polymer chain is one of the recent hot topics, because the dual metal species controlled in the polymer are expected to expand the functions of MSPs.^{2,8-17}

Main text (Page 13)

We revised the following sentences,

“It should be noted that if we look the chemical structure of the modified ditopic ligand TOSRuT closely, an asymmetric structure can be considered for this ligand due to presence of one Os(II) and one Ru(II) complex into the modified ditopic ligand. Therefore, the reactivity of two tpy units at the two ends of TOSRuT may differ little bit due to presence of two heterometal ions. In other way, the TOSRuT can be imagined as a structure with one side as head and another side as tail. Thus, when TOSRuT reacts with Fe(II) to make the polymer, it could be anticipated that the resultant polymer (polyOsRuFe) may be formed in regular head-to-tail structure (Fig. 2a) or regular structure with some irregular head-to-head/tail-to-tail structure (Fig. 2b). That means the sequence of Fe(II)/Ru(II)/Os(II) in polyOsRuFe may be repeated in alternate fashion or the sequence may be repeated with small irregularity.”

to

*“It should be noted that if we look the chemical structure of the modified ditopic ligand TOSRuT closely, an asymmetric structure can be considered for this ligand due to presence of one Os(II) and one Ru(II) complex into the modified ditopic ligand. **If the reactivity of two tpy units at the two ends of TOSRuT is different due to presence of two heterometal ions, the TOSRuT** can be imagined as a structure with one side as head and another side as tail. Thus, when TOSRuT reacts with Fe(II) to make the polymer, it could be anticipated that the resultant polymer (polyOsRuFe) may be formed in regular head-to-tail structure (Fig. 2a) or regular structure with some irregular head-to-head/tail-to-tail structure (Fig. 2b). **For example, if the Ru side binds Fe a lot better than the Os side, the first step of the assembly process would be the formation of the OsRuFeRuOs dimers which then would lead to an ordered (OsRuFeRuOs-Fe-OsRuFeRuOs)_n polymeric sequence. However, if the reactivity of two tpy units at the two ends of TOSRuT is same, the complexation with Fe would lead to polyOsRuFe with a complete random sequence.**”*

Conclusion (the first sentence):

“We succeeded in sequence control of three different transition metals in metallo-supramolecular polymer for the first time and achieved the quad-color electrochromism.” was changed to *“We succeeded in **stepwise introduction** of three different transition metals in metallo-supramolecular polymer for the first time and achieved the quad-color electrochromism.”*

Finally, thank you very much once again for taking precious time from your work and helping us in improving this manuscript. We have tried our best to address your concern. We sincerely hope that the current fully revised manuscript would meet your concern.

Our reply to Reviewer-3

Remark: *The authors carefully addressed all comments raised by me and all other referees. I would suggest publishing this manuscript in its current version.*

Our reply: Many thanks for your recommendation for publication.